# Visual Prompting Reimagined: The Power of Activation Prompts

## Abstract

Visual prompting (VP) has emerged as a popular method to repurpose pretrained vision models for adaptation to downstream tasks. Unlike conventional model fine-tuning techniques, VP introduces a universal perturbation directly into the input data to facilitate task-specific fine-tuning rather than modifying model parameters. However, there exists a noticeable performance gap between VP and conventional fine-tuning methods, highlighting an unexplored realm in theory and practice to understand and advance (input-level) VP to reduce its current performance gap. Towards this end, we introduce a generalized concept, termed activation prompt (AP), which extends the scope of (input-level) VP by enabling universal perturbations to be applied to activation maps within the intermediate layers of the model. By using AP to revisit the problem of VP and employing it as an analytical tool, we demonstrate the intrinsic limitations of VP in both performance and efficiency, revealing why input-level prompting may lack effectiveness compared to AP, which exhibits a model-dependent layer preference. We show that AP is closely related to normalization tuning in convolutional neural networks (CNNs) and vision transformers (ViTs), although each model type has distinct layer preferences for prompting. We also theoretically elucidate the rationale behind such preference by analyzing global features across layers. Through extensive experiments across 29 datasets and various model architectures, we provide a comprehensive performance analysis of AP, comparing it with VP and parameter-efficient fine-tuning (PEFT) baselines. Our results demonstrate AP's superiority in both accuracy and efficiency, considering factors such as time, parameters, memory usage, and throughput.

## 1 Introduction

Large pretrained models have emerged as fundamental components in deep learning (DL) (Brown et al., 2020; Touvron et al., 2023; Chiang et al., 2023; Li et al., 2022; Bai et al., 2023a) in recent years. Despite their exceptional performance, the substantial increase in computational demands, as highlighted in recent studies (Frantar and Alistarh, 2023), has underlined the need for more economical and lightweight fine-tuning approaches. Thus, the pretraining-finetuning paradigm rises, allowing for quickly adapting a pretrained model to a plethora of downstream tasks (Jia et al., 2022; Hu et al., 2021; Chen et al., 2022a; Cai et al., 2020; Sung et al., 2022; Pfeiffer et al., 2020; Chen et al., 2023a). Among the various parameter-efficient finetuning (PEFT) methods (Hu et al., 2021; Chen et al., 2022a; Pfeiffer et al., 2020; He et al., 2021; Xu et al., 2023), *prompting technique* has been gaining popularity in the vision domain (Liu et al., 2023; Li and Liang, 2021).

Different from the model-centric PEFT techniques in computer vision (CV), the conventional visual prompting (**VP**) crafts specific input perturbations (known as 'prompts') to reprogram the pretrained model for a targeted task, without altering the model parameters. This offers a new data-centric viewpoint to analyze, understand, and harness the pretrained model (Chen et al., 2023a). However, despite the recent advancement, the performance of state-of-the-art (SOTA) VP methods still lags behind model-based fine-tuning methods (Chen et al., 2023a; Wu et al., 2022). It appears that the potential of VP has not been fully realized for vision models, particularly when considering its relative progress compared to its counterpart in natural language processing (NLP) (Liu et al., 2023; Li and Liang, 2021). In this work, **we aim to** provide a rigorous and comprehensive examination of VP and explore its enhancement tailored for vision models, including convolutional neural networks (CNNs) and vision Transformers (ViTs). In particular, we ask:

> ***(Q)** Is VP (visual prompting) truly beneficial for improving vision models and tasks, and under what conditions does it prove effective or ineffective?*

To tackle question **(Q)**, we present a generalized variant of VP termed activation prompt (**AP**), which involves the incorporation of learnable perturbations into the activation maps of intermediate layers, rather than focusing solely on the input layer. See **Fig. 1** for an illustration. The introduction of AP allows us to study the (in)effectiveness of (input-level) VP, as VP can be treated as a specific realization of AP. By employing AP as both a bridge and an analytical tool, we show that the conventional input-based VP might not be the most effective or efficient design. In fact, appropriately implemented AP can outperform traditional VP significantly. To shed light on the underlying mechanism of AP, we present both empirical evidence and theoretical insights. It is also worth noting that, unlike VP, which can be applied in a black-box model setting (Tsai et al., 2020; Oh et al., 2023), AP requires modifying the parameters of intermediate activation maps and is only applicable in a white-box setting.

The work **most relevant** to ours is (Jia et al., 2022), which also integrates prompts with intermediate layers of ViTs, resulting in the method known as visual prompt tuning (VPT). However, our work has the following distinctions from VPT. *First*, AP and VPT diverge in their designs. AP concentrates on the targeted application of prompts to a single model layer. In contrast, VPT and its deep variant (termed VPT-deep) apply prompts across multiple layers. Specifically, VPT-deep initiates prompts at one layer and extends them across all subsequent layers. The distinctive layer-prompting approach makes VPT not covering VP as a special case. In con-

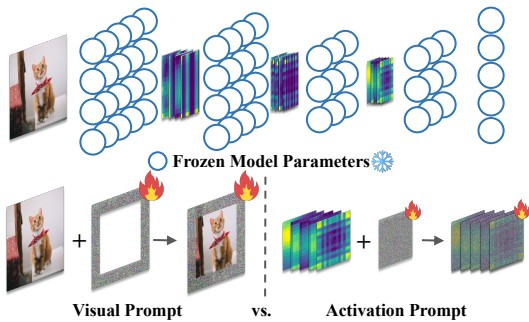

**Frozen Model Parameters** ❄

**Visual Prompt**     vs.     **Activation Prompt**

Figure 1: An illustration of the proposed activation prompt vs. the conventional input-based prompt.

trast, AP serves as a generalized framework for VP, making it easier to analyze its effectiveness. *Second*, this work identifies the layer preference of vision models regarding prompts. Through AP, we can gain insights into these layer preferences on both CNNs and ViTs. In contrast, VPT does not conduct a systematic analysis of layer and architectural type effects. *Third*, another notable difference between our work and the VPT study is our theoretical analysis. We establish a connection between AP and normalization tuning and theoretically validate the concept of layer preference and its influence on various architectural designs. Our theoretical analysis also shows that the traditional implementation of input-level VP could be suboptimal. In summary, our **contributions** include:

• We propose AP (activation prompt) as a valuable tool for gaining insights into VP (visual prompting). And AP establishes itself as a versatile and effective prompting technique in its own right, revealing a provable relationship with normalization tuning (Sec. 3).

• We offer an in-depth analysis of AP's layer preference and its architecture effects. Through empirical studies, we unveil the connection between the layer preference and the capacity for capturing global features (Sec. 4). In addition, we theoretically validate those findings (Sec. 5).

• Through extensive experimentation involving 29 datasets across various benchmarks, we affirm that AP enhances the input-level VP in diverse learning scenarios. Furthermore, AP narrows the performance gap even when compared to 6 other stateful PEFT methods.

## 2 RELATED WORK

**Visual prompting.** VP was first proposed in (Bahng et al., 2022a; Jia et al., 2022) to extend the prompting technique in NLP. A similar idea with a different name, known as adversarial reprogramming, was also proposed earlier in CV (Elsayed et al., 2018; Chen, 2022; Neekhara et al., 2018; 2022; Chen et al., 2021; Zhang et al., 2022a; Chen et al., 2022b). It aims at re-purposing a fixed pretrained model to adapt to a new task. Recent advancement focuses on improved label mapping (Chen et al., 2021; Yang et al., 2023) and normalization strategy (Wu et al., 2022) to enhance VP. Other works further extend VP to areas like adversarial defense (Chen et al., 2023b; Mao et al., 2022) and distribution shift (Huang et al., 2023a; Tsai et al., 2023), and vision-language models (Zhou et al., 2022).

**Theoretical study on prompt engineering.** Existing theoretical works on prompt engineering include the expressive power of the introduced parameter (Wei et al., 2021; Bai et al., 2023b; Akyürek et al., 2022), the optimization process (Ding et al., 2022; Von Oswald et al., 2023), and the generalization analysis (Xie et al., 2021; Oymak et al., 2023; Zhang et al., 2023a; Li et al., 2023a; Huang et al., 2023b; Li et al., 2024a;b). Most studies concentrate on in-context learning, a tuning-free hard prompt method. In contrast, for soft prompt tuning, Wei et al. (2021) show that prompting is powerful enough to remove nonessential information for the downstream task. Ding et al. (2022) interpret prompt tuning as a subspace optimization method for the solution or functional space. Notably, there is solely one study (Oymak et al., 2023) on the generalization dynamics of gradient-based prompt tuning but relying on a single-layer Transformer architecture without the MLP layer, making it incapable of examining the impact of multiple layers.

**Parameter-efficient fine-tuning (PEFT).** PEFT demonstrates that only finetuning a small part of a large pretrained model can achieve outstanding performance. In the domain of CV, besides prompting-based methods, PEFT methods can be roughly classified into two categories. The former (Basu et al., 2023; Xu et al., 2023) focuses on identifying a small ratio of parameters to update from the pretrained model, such as normalization tuning (Basu et al., 2023). The latter designs additional modules to the original network backbone to adapt to downstream tasks (Hu et al., 2021; Chen et al., 2022a; Pfeiffer et al., 2020; Xu et al., 2023; Karimi Mahabadi et al., 2021; Lian et al., 2022; Zhang et al., 2022b; Luo et al., 2023). Examples include LoRA (Hu et al., 2021), adapter-based methods (Chen et al., 2022a; Pfeiffer et al., 2020; Karimi Mahabadi et al., 2021; Luo et al., 2023), and FACT (Jie and Deng, 2023) that tensorizes the ViT weights to a 3D tensor and reduces the tunable parameter ratio to less than 0.01%. We note that AP differentiates itself from the methods above by avoiding additional inference overheads or any requirements on the model architectures.

## 3 ACTIVATION PROMPT: DESIGN AND RATIONALE

**Preliminaries on classical VP.** VP harnesses universal pixel-level perturbations applied to input images as a means of model adaptation (Bahng et al., 2022b). For example, VP enables the transfer learning of an ImageNet-trained source model to various downstream tasks without the need for fine-tuning the model weights. It has sparked significant interest in the recent research (Chen et al., 2023a; Wu et al., 2022; Zhang et al., 2022a; Bahng et al., 2022b; Tsai et al., 2020). Concretely, let $f_{\boldsymbol{\theta}}$ denote the pre-trained source model parameterized by $\boldsymbol{\theta}$, and $\mathcal{D} = \{(\boldsymbol{x}_1, y_1), (\boldsymbol{x}_2, y_2), \ldots, (\boldsymbol{x}_N, y_N)\}$ denote the fine-tuning dataset for a downstream task, with $\boldsymbol{x}$ and $y$ being the data feature and label, respectively. **The objective of VP** is to obtain a perturbation vector, denoted as $\boldsymbol{\delta}_{\mathrm{VP}}$, which is tailored to a specific task but remains agnostic to the input data. This vector is then used to transform the input data $\boldsymbol{x}$ through the function $g(\boldsymbol{x}, \boldsymbol{\delta}_{\mathrm{VP}})$. Here $g$ symbolizes the transformation template function that molds the input image to fit the desired prompt pattern. Two prevalent templates include the addition $g(\boldsymbol{x}, \boldsymbol{\delta}_{\mathrm{VP}}) = \boldsymbol{x} + \boldsymbol{\delta}_{\mathrm{VP}}$ (Zhang et al., 2022a; Bahng et al., 2022b), and the resize-and-concatenation $g(\boldsymbol{x}, \boldsymbol{\delta}_{\mathrm{VP}}) = [\boldsymbol{\delta}_{\mathrm{VP}}, M(\boldsymbol{x})]$ (Chen et al., 2023a; Zhang et al., 2022a), where $M$ is the resizing function. Unless specified otherwise, we consider the additive VP formulation.

**Activation prompt (AP): Generalizing VP in feature space.** The conventional VP approach primarily focuses on making direct modifications to the input data. However, this direct manipulation may have two limitations. *First*, raw input data typically contains an abundance of details, which can introduce complications for tasks like prompt generation due to issues such as background clutter and semantic ambiguity (Yu et al., 2017). In contrast, intermediate features tend to encompass a broader range of local and global attributes, preserving more class-discriminative information for decision-making (Bau et al., 2017). *Second*, parameter updates in VP demand gradient propagation throughout the entire network. Consequently, even with a lower number of tunable parameters, the training cost may increase.

Motivated by the above, we broaden the scope of VP into the feature domain and introduce the concept of **activation prompting (AP)**, see Fig. 1 for an illustration. Given a neural network model with $L$ layers, represented as $\boldsymbol{\theta} = [\boldsymbol{\theta}^{(1)}, \boldsymbol{\theta}^{(2)}, \ldots, \boldsymbol{\theta}^{(L)}]$, the output from the $l$-th layer is denoted as $\boldsymbol{z}^{(l)} = f_{\boldsymbol{\theta}^{(l)}}(\boldsymbol{z}^{(l-1)})$, where $\boldsymbol{z}^{(0)} = \boldsymbol{x}$ (*i.e.*, the input date). Similar to VP, AP at the $l$-th layer is defined by a perturbation vector $\boldsymbol{\delta}^{(l)}$ to the intermediate feature $\boldsymbol{z}^{(l)}$, leading to the 'prompted' feature map $g(\boldsymbol{z}^{(l)}, \boldsymbol{\delta}^{(l)}) = \boldsymbol{z}^{(l)} + \boldsymbol{\delta}^{(l)}$. We denote the output with the $l$-th-layer AP given $\boldsymbol{\theta}$ as $f_{\boldsymbol{\theta}}(\boldsymbol{x}, \boldsymbol{\delta}^{(l)})$. **The objective of AP** is then to optimize $\boldsymbol{\delta}^{(l)}$ so as to facilitate the adaptation of the fixed source

model $f_{\boldsymbol{\theta}}$ for performing the downstream task on $\mathcal{D}$. It is evident that AP can be conceptualized as an extension of VP when we set the layer number $l$ to 0. Moreover, the optimization process for both VP and AP can be carried out similarly through empirical risk minimization (ERM) on $\mathcal{D}$, *i.e.*, $\min_{\boldsymbol{\delta}^{(l)}} \frac{1}{|\mathcal{D}|} \sum_{(\boldsymbol{x},y) \in \mathcal{D}} \ell(f_{\boldsymbol{\theta}}(\boldsymbol{x}, \boldsymbol{\delta}^{(l)}); y)$, where $\ell$ is the sample-wise cross-entropy loss.

AP also exhibits several notable attributes different from VP. *First*, the number of parameters in AP directly relates to the size of the feature map $\boldsymbol{z}^{(l)}$. Hence, a properly designed AP can substantially reduce the parameter count. *Second*, while the optimization of AP mirrors that of VP, its parameter update does not necessitate back-propagation throughout the entire network. For example, embedding AP deeper within the architecture reduces computational demands during training.

**AP could be a better design than VP.** Next, we present a preliminary experiment that serves as a *warm-up*, demonstrating how AP exhibits the potential to improve accuracy performance, as well as enhance computation and parameter efficiency when compared to VP. We examine the commonly used transfer learning scenario for applying VP, in which the source model ResNet-101 (He et al., 2016) is initially trained on ImageNet (Deng et al., 2009) and is subsequently transferred to the CIFAR-10 dataset (Krizhevsky et al., 2009). **Fig.** 2 presents a performance comparison between AP and VP against the layer index on ResNet-101, at which AP is introduced. The preliminary results provide several key insights, which will be substantiated in more detail later. *First*, AP holds the potential to substantially enhance the accuracy of transfer learning when compared to VP. For instance, when AP is applied at layer 31, it

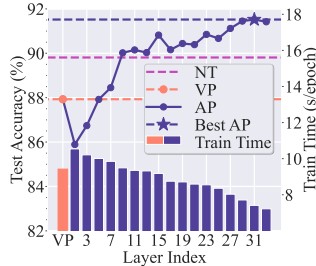

Figure 2: Performance and efficiency comparison of VP, NORM-TUNE and AP over different layers of ResNet-101 on OxfordPets.

achieves the highest accuracy in transfer learning, surpassing VP by approximately 5%. In fact, more comprehensive experiments presented in Sec. 6 demonstrate that applying AP to a *deeper* layer consistently produces the most significant accuracy improvements across a wide range of CNNs. *Second*, due to the preference for deeper layers when utilizing AP in CNNs, there exists a computational advantage since back-propagation from the output to the input layer is *not* required. *Third*, AP maintains the parameter efficiency merit compared to VP. For instance, at the layer that exhibits the best performance, AP utilizes only $100k$ parameters, whereas VP employs $150k$ parameters. The results from the warm-up experiment above indicate that *AP has the potential to outperform VP, offering not only improved accuracy but also greater efficiency*.

**Understanding AP through its connection to normalization tuning.** Normalization tuning (NORM-TUNE), as a PEFT technique, finetunes parameters within model's normalization layers, *i.e.*, BatchNorm for CNNs (Ioffe and Szegedy, 2015) and LayerNorm for ViTs (Ba et al., 2016). For clarity, we denote the tunable parameters of a normalization layer by $\boldsymbol{\gamma} = (\gamma_1, \cdots, \gamma_{D'})^{\top}$ for linear coefficients and $\boldsymbol{\beta} = (\beta_1, \cdots, \beta_{D'})^{\top}$ for biases, with $D'$ representing the number of channels or the token dimension. Further, define $\boldsymbol{\mu}$ and $\boldsymbol{\sigma}$ as the channel-wise mean and standard deviation constants of $\boldsymbol{z}^{(l)}$ for BatchNorm over the entire batch. For LayerNorm, they represent the data-wise mean and standard deviation of $\boldsymbol{z}^{(l)}$ across the embedding dimension. Given that both AP and NORM-TUNE utilize a linear model for feature representations, *i.e.*, $g(\boldsymbol{z}^{(l)}, \boldsymbol{\delta}^{(l)}) = \boldsymbol{z}^{(l)} + \boldsymbol{\delta}^{(l)}$ for AP and

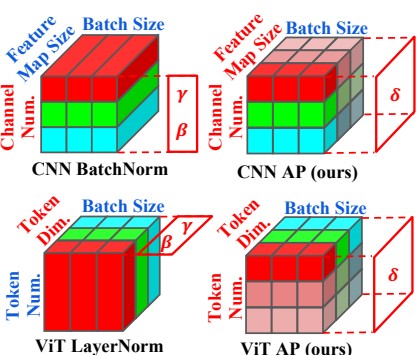

Figure 3: Tunable parameter shape comparison between NORM-TUNE and AP (ours). The same color indicates shared parameters across different dimensions.

$g(\boldsymbol{z}^{(l)}, \boldsymbol{\gamma}, \boldsymbol{\beta}) = \boldsymbol{\gamma} \cdot (\boldsymbol{z}^{(l)} - \boldsymbol{\mu}) / \sqrt{\boldsymbol{\sigma}} + \boldsymbol{\beta}$ for NORM-TUNE, AP can be interpreted as a variant of NORM-TUNE. **Fig.** 3 illustrates the connection; see elaboration below.

• *CNNs*: When AP's perturbations are consistent across all feature map units, the unit-scaling BatchNorm-based NORM-TUNE closely mirrors the formulation of AP, differentiated merely by a linear mapping plus a bias. This equivalence becomes apparent when relating $\boldsymbol{W}^{(l)} \boldsymbol{\delta}^{(l)}$ to $\boldsymbol{\beta} - \boldsymbol{\gamma} \cdot \boldsymbol{\mu} / \sqrt{\boldsymbol{\sigma}}$, especially when $\boldsymbol{\gamma} / \sqrt{\boldsymbol{\sigma}} = 1$, supposing $\boldsymbol{W}^{(l)}$ as the weight for the $l$-th layer.

• *ViTs*: Assuming uniform perturbations across tokens and consistent mean value across data dimensions within a batch, AP reduces to the unit-scaling LayerNorm-based NORM-TUNE. This can be represented as $\delta^{(l)} = \beta - \mu$, given $\gamma/\sqrt{\sigma} = 1$.

Due to more flexible perturbations of AP, such a connection exhibits increased power of AP than NORM-TUNE. We formally prove and summarize the proposed connection in Proposition 1 in Appx. C.2. Meanwhile, we remark that another key difference of AP compared to NORM-TUNE is that no parameters of the model backbone need to be altered during training. This differentiates "prompting" from other PEFT methods, where the former keeps the pretrained model backbone intact. In the realm of PEFT, recent research has also shown that LayerNorm-based NORM-TUNE serves as a robust baseline of model adaptation for ViTs (Basu et al., 2023). Beyond that, we will show that AP can surpass NORM-TUNE and remain effective for CNNs.

## 4 A DEEP DIVE INTO AP: LAYER AND ARCHITECTURE EFFECTS

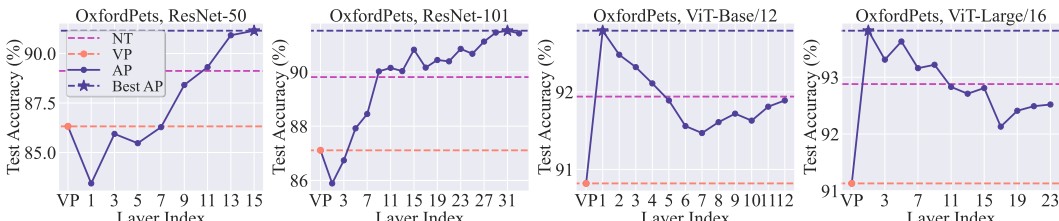

Figure 4: Layer preference of AP with different model architectures on OxfordPets (Parkhi et al., 2012). CNNs and ViTs exhibit opposite layer preferences. Results on more datasets are provided in Fig. A1.

Our preliminary findings in Fig. 2 suggest that the effectiveness of AP may be contingent on the *specific layer* where it is installed. To acquire a deeper understanding of this characteristic and its association with *model architecture*, we examine both ResNet and ViT model types.

**Fig. 4** follows and expands Fig. 2 by covering the additional models, *i.e.*, ResNet-50, ViT-Base/12, and ViT-Large/16, and showcasing the transfer learning accuracy enabled by AP on the downstream dataset OxfordPets as a function of the layer index to which AP is applied. As we can see, a key observation is that *ResNets and ViTs exhibit contrasting layer preferences for* AP, where ★ indicates the best performance of AP in Fig. 4 under each architecture. Specifically, CNNs exhibit a preference for AP in their *deeper* layers, while ViTs tend to favor AP in their *shallower* layers. Moreover, within the comfort layer zone, the performance of AP consistently outperforms NORM-TUNE.

**Dissecting CNNs and ViTs: AP prioritizes 'global' features over 'local' features.** To unpack the intriguing AP's layer preference behavior above, we next examine the features captured by different layers of CNNs and ViTs. To this end, we first employ the Centered Kernel Alignment (CKA)-based feature similarity analysis (Cortes et al., 2012) to measure the layer-wise representation similarity between CNNs and ViTs, *e.g.*, ResNet-101 and ViT-Large/16 in **Fig. 5**. As we

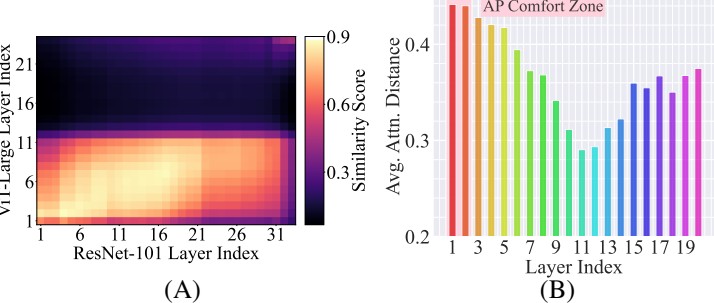

Figure 5: Features dissection to understand the layer effect of AP on OxfordPets dataset. (A) CKA-based feature similarity comparison between ViT-Large/16 and ResNet-101. (B) The average attention distance across all the heads of different layers of ViT-Large/16. A larger distance signifies a more globally-focused attention, indicative of global features.

can see, the deep features of ResNet-101 predominantly align with the middle layers of ViT-Large/16. This concurs with the observations made in (Raghu et al., 2021), which suggest that ViTs have the capability to capture features reminiscent of the deeper layers of CNNs even within their relatively early layers. In addition, as indicated by network dissection analysis for CNNs (Bau et al., 2017), it is known that CNNs tend to prioritize low-level visual concepts, *i.e.*, *local features* like color and texture, in their shallower layers. In contrast, they transition to high-level, class-discriminative concepts, encompassing *global features* like scenes and objects in deeper layers.

Drawing upon the analyses presented above and insights in **Fig. 4**, *we hypothesize* that AP exhibits a preference for deep layers in CNNs and shallow layers in ViTs, which can be attributed to the models' inclinations toward global features over local features. To bolster our hypothesis, we investigate how global information is distributed across the layers of ViTs. We employ a methodology used in (Raghu et al., 2021) and (Walmer et al., 2023) to compute the average attention distance between the position of query tokens and the locations they attend to with the query within each self-attention head in ViTs. This analysis unveils how each self-attention layer contributes to the balance between local and global information in the overall representation. In **Fig. 5** (B), we present the average attention distance across 16 attention heads for with different layer indices of a pretrained ViT-Large/16. A general trend can be observed: the distribution of the sorted attention distance moves firstly downwards (layer index from 1 to layer 12). This implies that the ratio of the global features captured by attention in general decreases. When the layer index is larger than 15, the global feature ratio slightly increases. This trend roughly aligns well with the patterns observed in Fig. 4. These observations underscore our claim that AP's layer preference is influenced by the presence of global features. We provide theoretical support in the following section to support the layer and architecture effect. In particular, we focus on the more challenging part of ViTs, since the study on CNNs is abundant. Furthermore, we provide theoretical support in the following section to support the layer and architecture effect.

**Remark on the comparison of AP vs. VPT**. While VPT (Jia et al., 2022) also suggests adding extra tokens (prompts) to all intermediate layers of a ViT, our approach differs fundamentally. AP was motivated to introduce a broader framework for VP, where prompts are applied to intermediate activations at any *single* layer, rather than across multiple or all layers as in VPT. This allows us to rigorously explore optimal layer selection for effective prompting, where (input-level) VP is covered as a special case. Unlike VPT, AP uncovers new insights into layer-specific effects, architectural dependencies, and their explanations, supported by both empirical and theoretical analyses (as will be evident later). Furthermore, our findings show that strategic layer selection in AP can match or surpass the effectiveness of VPT's multi-layer prompting (See Tab. 4 in Sec. 6).

## 5    THEORETICAL ANALYSES FOR LAYER AND ARCHITECTURE EFFECTS

From a perspective of generalization, we focus on studying the layer and architecture effect for **ViTs**: *To achieve the desired generalization performance (or test accuracy), will shallow-layer* AP *tuning require less sample complexity than deep-layer ones for ViTs?* If so, with the same sample complexity, shallow-layer AP could achieve better performance than deep-layer ones. To show this, we present the theoretical setups that satisfy the conditions of global features for ViTs, followed by the generalization analysis with sample complexity bound in Theorem 1.

**Problem setup.** Building on the theoretical frameworks for analyzing the training and generalization of Transformers (Li et al., 2023b; Oymak et al., 2023; Tarzanagh et al., 2023), we derive theoretical insights by considering a binary classification problem. We use a single-head, two-layer ViT (Huang et al., 2023c; Tian et al., 2023; Nichani et al., 2024; Li et al., 2023b) as the pretrained model, applied to the dataset $\{x_n, y_n\}_{n=1}^N$. Here $y_n \in \{+1, -1\}$, and each data $x_n \in \mathbb{R}^{d \times P}$ consists of $P$ tokens. The training is implemented by a mini-batch stochastic gradient descent (SGD) with the loss $\ell(f_{\boldsymbol{\theta}}(x_n, \boldsymbol{\delta}); y_n)$, where $f_{\boldsymbol{\theta}}$ and $\boldsymbol{\delta}$ are the pretrained model and the trainable AP, respectively. The generalization performance is evaluated by the population risk $\mathbb{E}[\ell(f_{\boldsymbol{\theta}}(x, \boldsymbol{\delta}); y)]$.

**Data assumption.** Each token of $x_n$ is formulated as a pattern added with a Gaussian noise following $\mathcal{N}(0, \sigma^2)$, $\sigma \leq O(1/P)$. We consider four patterns $\{v_1, v_2, v_3, v_4\}$ in total. In each $x_n$, only one token corresponds to either $v_1$ or $v_2$, named discriminative patterns that decide the label. Other $P - 1$ tokens correspond to either $v_3$ or $v_4$, named non-discriminative patterns that are irrelevant ones for the downstream task. For instance, if one token within $x_n$ is the noisy version of $v_1$ ($v_2$), then its corresponding downstream task label $y^n = 1$ ($y^n = -1$).

**Pretrained model assumption.** We have mild assumptions on the MLP neuron weights and self-attention matrices of the pretrained model, which have been used in existing works or verified in numerical experiments. Specifically, recent SOTA theoretical findings (Shi et al., 2022; Li et al., 2023b; Wen and Li, 2021) reveal, during the pretraining stage, the weights of each neuron in the MLP tend to converge towards one of the patterns present in the raw data, e.g, $v_1, v_3$. Following the observation above, we assume neuron weights in the $\ell$-th MLP after pretraining to be one of the patterns in $\{v_1, v_2, v_3, v_4\}$. Typically, $v_1$ and $v_2$ are patterns observed in the downstream task that

have relevance to the labels, while $v_3$ and $v_4$ are patterns also present in the downstream task but do not bear a relation to the labels. In addition, as suggested by the global features introduced in Section 4 that make tokens attend to other tokens, we assume the key and value matrices to be scalings of permutation matrices. The details about the data and model assumptions can be found in Appx. C.3.

Given a set of queries $q_1, \cdots, q_P$ and keys $k_1, \cdots, k_P$ for an attention head, we formally define the *average attention distance* mentioned in **Fig. 5** as $\sum_{i=1}^{P} |i - \arg\max_{j \in [P]} \langle k_j, q_i \rangle|/P$, i.e., the average distance between the query $q_i$ and the key $k_j$ that has the largest inner product with $q_i$, $i, j \in [P]$. Assuming the discriminative key and value are away from the discriminative query with a distance of $d_A \geq 1$, we have the following Lemma on decreasing the average attention distance.

**Lemma 1** *The average attention distance defined above decreases from $(1 + d_A)/P$ to $1/P$ after the 1st layer of the simplified two-layer ViT.*

Lemma 1 supports our empirical observation in **Fig. 5 (B)** of decreasing attention distance values within deep layers in ViT. In addition, the reduction in the attention distance leads to an increased sample complexity, as summarized in the following theorem.

**Theorem 1** *Training a two-layer ViT with SGD returns a model with zero generalization error, as long as the batch size $B \geq \Omega(1)$, and the required number of samples $N$ satisfy either (i) $N \geq N_1 = \Theta(P)$ if adding AP to the 1st layer; (ii) $N \geq N_2 = \Theta(P^2 \log P)$ if adding AP to the 2nd layer. $N_2$ is order-wise larger than $N_1$.*

Theorem 1 shows deep-layer AP requires more training samples than the shallow one to achieve the same generalization, as shown by the dashed line in **Fig. 6**. Accordingly, with the same number of training samples and setup, shallow-layer AP generalizes better. The proof of Theorem 1 can be found in Sec. C.4. The basic proof idea is that for AP in the shallow layer, a trained prompt with a norm of $\Theta(P)$ that removes non-discriminative patterns is enough to make all tokens attend to discriminative tokens. Thus, the amount of global features does not decrease. This can ensure zero generalization by abundant global features. For AP in deep layers, however, given Lemma. 1, a lack of global features leads to an evident mismatch between discriminative tokens in the 2nd-layer self-attention. Hence, a trained prompt with a norm of $\Theta(P^2 \log P)$ is necessary to direct the attention to focus on discriminative tokens. The proof concludes with the demonstration that the sample complexity bound is proportional to the the trained prompts magnitude.

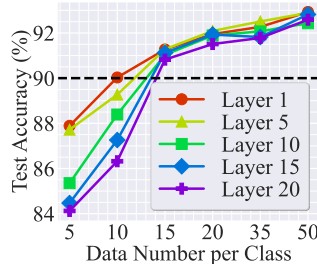

Figure 6: Sample complexity study of AP in different layers on OxfordPets with ViT-Large/16.

## 6 EXPERIMENTS

### 6.1 EXPERIMENT SETUP

**Datasets and models.** We utilize two commonly used architectures for the source datasets: ResNet-101 from the ResNet family (He et al., 2016) and ViT-Large/16 from the ViT family (Dosovitskiy et al., 2020). Both are pretrained on ImageNet-1K (Russakovsky et al., 2015). In the target domain, we consider over 20 datasets from transfer learning benchmarks FGVC (Maji et al., 2013) and VTAB (Zhai et al., 2019). In VTAB, we consider both *full-data* and *few-shot* (VTAB-1k) regimes. In addition, we also consider other commonly used datasets (Chen et al., 2023a) for transfer learning like CIFAR-10 (Krizhevsky et al., 2009), UCF101 (Soomro et al., 2012), GTSRB (Houben et al., 2013), Food101 (Bossard et al., 2014), and Waterbirds (Sagawa et al., 2019). More details on the datasets and the benchmarks can be found in Appx. A.

We cover three types of baselines in transfer learning. *First*, we primarily compare AP to finetuning methods designed for both CNNs and ViTs in transfer learning. These include LINEAR-PROBE that only finetunes the classification head with a fixed feature extractor, the conventional (input-level) VP (Bahng et al., 2022b) and NORM-TUNE (Basu et al., 2023) that tunes *all* the normalization layers in a model. *Second*, we select FULL-FINETUNE as our reference method due to its superior accuracy, which fine-tunes the entire pretrained model, albeit being the most computationally expensive option. *Third*, we consider other 9 SOTA PEFT baselines used in ViTs: VPT (Jia et al., 2022), GATEVPT (Yoo et al., 2023), E2VPT (Han et al., 2023), LoRA (Hu et al., 2021), ADAPTER (Chen

Table 1: Performance comparison of various methods on 19 datasets from different benchmarks. Three parameter-efficient baselines (denoted by ○) are compared to AP due to their high relevance, where the best performance is highlighted in **bold**. The most computationally intensive FULL-FINETUNE (denoted by ●) serves as the performance reference. Each accuracy value is averaged over 5 independent trials, with the variance omitted due to its negligible values ($\leq 0.3\%$). The "Average" column represents the averaged accuracy of each method over all the datasets in each row.

| Benchmark | | FGVC | | | | | VTAB | | | | | | | | | Others | | | | | Average |
|---|---|---|---|---|---|---|---|---|---|---|---|---|---|---|---|---|---|---|---|---|---|
| Architecture | | CUB200 | StanfordDog | StanfordCars | NA-Birds | OxfordFlowers | CIFAR-100 | Caltech-101 | DTD | Flowers102 | OxfordPets | SVHN | SUN397 | Camelyon | EuroSAT | CIFAR-10 | GTSRB | UCF101 | Food101 | Waterbirds | |
| **ResNet-101** ● FULL-FINETUNE | | 88.91 | 90.13 | 87.76 | 84.45 | 99.98 | 92.24 | 99.13 | 79.97 | 99.81 | 90.49 | 97.14 | 79.19 | 91.13 | 99.13 | 97.24 | 97.68 | 88.32 | 82.72 | 96.69 | 91.69 |
| ○ LINEAR-PROBE | | 63.76 | 86.63 | 49.62 | 52.09 | 82.01 | 73.87 | 90.58 | 61.35 | 93.14 | 91.17 | 66.30 | 54.51 | 83.36 | 95.84 | 92.25 | 79.64 | 71.03 | 64.31 | 88.11 | 75.76 |
| ○ NORM-TUNE | | 66.39 | 87.59 | **67.64** | 56.72 | 66.50 | **82.58** | 91.32 | 63.53 | 92.85 | 89.81 | **95.26** | 54.56 | 84.42 | 96.14 | 93.90 | **96.43** | 69.44 | **72.54** | **88.95** | 79.81 |
| ○ VP | | 65.72 | 86.91 | 51.04 | 54.23 | 78.50 | 72.01 | 93.51 | 63.12 | 90.17 | 87.93 | 80.68 | 54.97 | 83.71 | 95.44 | 92.55 | 83.18 | 66.30 | 57.89 | 86.71 | 76.03 |
| ○ AP (ours) | | **69.42** | **87.79** | 59.06 | **58.31** | **85.14** | 76.94 | **94.85** | **69.80** | **95.13** | **91.31** | 87.30 | **56.83** | **84.91** | **97.21** | **94.08** | 90.43 | **73.96** | 68.12 | 88.13 | **80.45** |
| **ViT-Large/16** ● FULL-FINETUNE | | 89.79 | 93.31 | 89.42 | 84.75 | 99.91 | 93.19 | 99.25 | 75.30 | 99.39 | 93.35 | 98.13 | 79.31 | 91.93 | 97.92 | 98.30 | 97.90 | 89.25 | 86.16 | 97.93 | 92.34 |
| ○ LINEAR-PROBE | | 84.69 | 86.11 | 65.24 | 75.71 | 99.40 | 88.55 | 97.01 | 73.31 | 99.24 | 91.15 | 65.79 | 72.37 | 84.05 | 97.26 | 98.13 | 80.72 | 83.02 | 83.02 | 94.16 | 85.20 |
| ○ NORM-TUNE | | 85.90 | 89.76 | **75.61** | 78.78 | 99.35 | 90.69 | 98.01 | 78.90 | 99.76 | 92.88 | 88.30 | 73.57 | 79.82 | 97.17 | 98.44 | 90.86 | 85.15 | 83.21 | 94.36 | 88.45 |
| ○ VP | | 85.24 | 87.02 | 67.64 | 76.20 | 99.32 | 89.44 | 97.81 | 77.72 | 99.72 | 91.31 | 85.70 | 74.33 | 84.27 | 97.85 | **98.80** | 89.09 | 84.67 | 82.23 | **95.03** | 87.54 |
| ○ AP (ours) | | **86.74** | **90.83** | 69.41 | **79.83** | **99.70** | **90.96** | **98.99** | 78.96 | **99.84** | **93.89** | **88.87** | **75.44** | **86.99** | **98.33** | 98.54 | **91.49** | **86.80** | **84.04** | 94.60 | **89.17** |

et al., 2022a), BIAS (Zaken et al., 2021), NORM-TUNE (Basu et al., 2023), ATTNSCALE (Basu et al., 2023), ADAPTERFORMER (Chen et al., 2022a), and SSF (Lian et al., 2022).

**Implementation, training, and evaluations.** We implement AP at the input of the third-to-last ResNet block in ResNet-101 and the third Transformer block in ViT-Large/16, based on the layer effect in Fig. 4. During training, all the methods are trained for 100 epochs using the Cross-Entropy loss with an Adam optimizer (Kingma and Ba, 2015). Hyperparameters, including learning rates, are determined through a search process for each method; see implementation details in Appx. A. During evaluation, we compare different methods in terms of their performance (testing accuracy) and efficiency. In particular, we depict the **efficiency portrait** of a method from the following 4 different perspectives: (1) tunable parameter number, (2) memory cost, (3) train time per epoch, and (4) throughput for inference efficiency, as will be shown in Tab. 2.

## 6.2 EXPERIMENT RESULTS

**AP is not only effective but also efficient.** We examine the performance of the proposed AP in the full-data regime below. *Two key observations* can be drawn from experiment results: (1) AP consistently outperforms baselines across the majority of datasets, in particular with a significant improvement over VP (Tab. 1); (2) AP demonstrates remarkable efficiency across various efficiency metrics, establishing itself as a cost-effective method (Tab. 2).

**Tab. 1** shows the performance of AP vs. the baselines: VP, NORM-TUNE, LINEAR-PROBE, and FULL-FINETUNE. As we can see, AP consistently outperforms VP in *all* the 19 datasets. Notably, AP yields an increase in the average accuracy of over 4% and 1.5% compared to VP for both ResNet-101 and ViT-Large/16. In some datasets, such as StanfordCars, SVHN and GT-SRB using ResNet-101, this advantage can increase to 7%~9%. AP also remains effective compared to NORM-TUNE, which has proven to be a strong PEFT method for ViT families in Basu et al. (2023). AP performs the best in 13 and 15 out of 19 datasets for ResNet-101 and ViT-Large/16, respectively. Although FULL-FINETUNE remains the best-performing in most

Table 2: An overview of the methods considered in this work. The efficiency analysis is based on the model-data setting (ViT-Large, CIFAR-10) with a batch size of 128, and time consumption is evaluated using a single RTX-A6000 GPU. For each metric, we use ↑ or ↓ to indicate whether a larger smaller value is favored for each metric.

| Method | Param. Efficiency Parameter # (M) ↓ | Train-Time Efficiency | | |
|---|---|---|---|---|
| | | Memory Cost (G) ↓ | Time Cost (s/epoch) ↓ | Troughput (image/s) ↑ |
| **ResNet-101** | | | | |
| FULL-FINETUNE | 44.5 | 10.32 | 118 | 41.47 |
| LINEAR-PROBE | 0.02 | 6.2 | 39 | 41.33 |
| NORM-TUNE | 0.13 | 11.7 | 83 | **41.45** |
| VP | **0.12** | 12.2 | 72 | 40.59 |
| AP | **0.12** | **6.3** | **41** | 41.36 |
| **ViT-Large/16** | | | | |
| FULL-FINETUNE | 304.33 | 41.5 | 520 | 79.58 |
| LINEAR-PROBE | 0.01 | 9.7 | 121 | 79.64 |
| NORM-TUNE | **0.06** | **29.5** | 285 | **79.51** |
| VP | 0.11 | 35.9 | 280 | 77.14 |
| AP | 0.16 | 31.6 | **262** | 79.48 |

datasets, AP still manages to approach and surpass it; see OxfordPets for ResNet-101 and DTD for ViT-Large/16. Importantly, AP is much more efficient than FULL-FINETUNE, as illustrated below.

Table 3: Performance comparison of various methods in the few-shot setting on the VTAB-1$K$ benchmark. Other settings follow Tab. 1.

| Architecture | Benchmark | VTAB-Natural | | | | | | | VTAB-Specialized | | | | VTAB-Structured | | | | | | | | Average |
|---|---|---|---|---|---|---|---|---|---|---|---|---|---|---|---|---|---|---|---|---|---|
| | | Caltech101 | CIFAR-100 | DTD | Flowers102 | OxfordPets | Sun397 | SVHN | Camelyon | EuroSAT | Resisc45 | Retinopathy | Clevr-Count | Clevr-Dist | DMLab | dSpr-Loc | dSpr-Ori | KITTI-Dist | sNORB-Azim | sNORB-Elev | |
| ResNet-101 | ● Full-Finetune | 89.99 | 45.17 | 63.78 | 84.29 | 89.82 | 41.09 | 67.79 | 84.92 | 74.57 | 91.37 | 74.14 | 58.11 | 60.99 | 43.61 | 67.05 | 40.45 | 78.34 | 33.64 | 36.38 | 64.50 |
| | ○ Linear-Probe | 83.87 | 39.13 | 53.09 | 70.89 | 85.15 | 28.14 | 43.44 | 78.65 | 69.43 | 90.78 | 69.31 | 35.91 | 36.48 | 35.75 | 34.76 | 19.51 | 65.68 | 16.91 | 23.39 | 51.12 |
| | ○ Norm-Tune | 85.61 | 35.78 | 47.71 | 56.64 | 78.10 | 10.10 | **68.67** | **83.16** | 61.10 | 90.50 | **72.44** | 37.54 | 55.24 | **40.04** | **60.89** | 20.33 | 65.54 | 24.86 | 25.96 | 53.70 |
| | ○ VP | 84.73 | **43.01** | 57.55 | 76.91 | 87.03 | 28.75 | 55.47 | 75.15 | 70.27 | 89.26 | 69.08 | 36.70 | 54.24 | 34.48 | 42.41 | 20.32 | 63.71 | 17.93 | 26.93 | 54.42 |
| | ○ AP | **87.49** | 39.80 | **63.62** | **81.44** | **88.74** | **34.83** | 65.92 | 78.91 | **74.19** | **91.44** | 71.18 | **40.20** | **55.26** | 38.95 | 54.68 | **21.98** | **72.86** | **26.24** | **28.77** | **58.76** |
| ViT-Large/16 | ● Full-Finetune | 93.34 | 76.03 | 75.74 | 99.88 | 93.72 | 59.06 | 68.70 | 86.70 | 82.84 | 93.54 | 82.22 | 55.42 | 60.33 | 48.23 | 83.62 | 52.77 | 78.06 | 30.40 | 29.95 | 71.08 |
| | ○ Linear-Probe | 89.37 | 62.98 | 70.02 | 93.42 | 91.22 | 53.68 | 45.28 | 80.52 | 80.34 | 91.64 | 70.43 | 38.15 | 35.26 | 40.74 | 21.84 | 29.42 | 62.54 | 14.59 | 23.09 | 57.60 |
| | ○ Norm-Tune | 91.10 | **65.20** | 72.36 | 98.64 | 91.38 | 55.14 | 47.21 | **82.50** | 82.34 | **93.94** | 71.74 | **44.23** | 44.59 | **41.21** | 35.64 | **32.08** | 63.43 | 16.52 | 24.12 | 60.68 |
| | ○ VP | 90.06 | 63.16 | 71.59 | 95.35 | 91.20 | 54.45 | 46.26 | 81.82 | 81.45 | 92.25 | 71.03 | 41.03 | **45.49** | 39.94 | 32.52 | 30.29 | 62.68 | 15.59 | 23.13 | 59.96 |
| | ○ AP | **91.40** | 64.40 | **72.61** | **99.50** | **91.46** | **56.67** | **49.43** | 81.41 | **82.76** | 93.14 | 71.99 | 43.26 | 38.09 | 40.57 | **42.44** | 31.83 | **65.40** | **18.29** | **25.96** | **61.06** |

**Tab. 2** demonstrates the efficiency profile of different methods under different metrics. Two key insights can be drawn from the results. *First*, in comparison to VP, AP demonstrates superior efficiency in terms of memory (reduced memory overhead), time (decreased training duration), and inference (increased throughput) for both ResNet-101 and ViT-Large/16. This superiority is maintained while operating at a comparable parameter efficiency, marked by a negligible tunable ratio difference of less than $0.05\%$. This trend is amplified for ResNet-101, as evidenced by the significant reductions in memory usage (6.3 G for AP vs. 12.2 G for VP) and training duration (41 s/epoch for AP vs. 72 s/epoch for VP). This efficiency arises from the AP's preference towards deeper layers over shallower ones in ResNet-101, resulting in reduced back-propagation overhead for most of the network. *Second*, when compared to Norm-Tune, although AP consumes slightly higher memory cost for ViT-Large/16, it achieves higher training efficiency for ResNet-101 and ViT-Large/16. This is due to that, while Norm-Tune possesses a small tunable parameter ratio, these parameters are dispersed throughout the network, leading to a more expensive back-propagation process. Although no significant difference is observed in throughput, we will show later in Tab. 4 that AP enjoys high throughput efficiency compared to other PEFT methods.

**How does the downstream dataset scale affect AP?** To study the effect brought by the downstream data scales, we follow the setting of Jia et al. (2022) and examine the performance of different methods under the few-shot setting on VTAB-1$K$. In particular, for each of the 19 datasets in the VTAB benchmark, only 1000 data samples are available for training. **Tab. 3** shows that AP makes a distinguishable improvement over the baselines VP and Norm-Tune in the few-shot setting. As we can see, AP achieves a performance boost of over $1\%$ than VP using ViT-Large/16 and this advantage increases to $4.3\%$ in the case of ResNet-101. This demonstrates that directly steering the intermediate features can be more effective when facing data scarcity.

**Comparing AP with VPT and more PEFT baselines.** As VP is introduced as a generalization of the conventional (input-level) AP, we do not anticipate it to outperform all model-based PEFT methods. Yet, to demonstrate its potential, **Tab. 4** compares the performance of AP with that of PEFT baselines, in particular with VPT (Jia et al., 2022). As we can see, even when compared to the stateful PEFT methods, AP still yields competitive performance in terms of both accuracy and efficiency. For example, AP ranks roughly 2∼4 in terms of accuracy among the 8 PEFT methods considered in this work. In

Table 4: Performance of AP and more SOTA PEFT methods on ViT-Large/16. Settings follow Tab. 1.

| | Accuracy | | | Efficiency | | | |
|---|---|---|---|---|---|---|---|
| | **Full-Data** | | | | **Train-Time Efficiency** | | |
| | FGVC | VTAB | Others | Param. # | Memory | Time | Throughput |
| Number of tasks | 5 | 9 | 5 | - | - | - | - |
| Full-Finetune | 91.43 | 91.97 | 93.91 | 304.33 | 41.5 | 520 | 79.58 |
| Linear-Probe | 82.23 | 78.90 | 87.81 | 0.01 | 9.7 | 121 | 79.64 |
| Bias | 85.32 | 89.84 | 90.41 | 0.29 | 32.9 | 297 | **79.43** |
| LoRA | 86.87 | 89.81 | 91.45 | 1.00 | 33.1 | 363 | **79.43** |
| VPT | 86.34 | 89.24 | 90.14 | 0.25 | 33.7 | 334 | 76.35 |
| GateVPT | 86.31 | 89.14 | 91.11 | 3.14 | 34.9 | 395 | 61.34 |
| E2VPT | **89.93** | 90.12 | 91.45 | 1.21 | 33.4 | 369 | 52.32 |
| Adapter | 87.06 | 89.44 | 91.21 | 2.17 | 32.4 | 357 | 63.39 |
| AdapterFormer | 89.18 | **90.69** | **92.08** | 0.65 | 32.3 | 289 | 23.69 |
| SSF | 87.32 | 89.43 | 92.21 | 0.48 | 34.7 | 299 | **79.49** |
| AP(Ours) | 85.30 | 90.25 | 91.09 | **0.16** | **31.6** | **262** | 79.43 |

addition, AP ranks the first from the efficiency perspective. In contrast, the best accuracy performance of AdapterFormer comes at a cost of three times lower throughput efficiency. This is due to that extra modules introduce significantly more computations during the inference.

**Applying AP to various model architectures.** To ensure that our conclusions generalize well, we shift our focus from the vision source model to the vision-language model, specific to CLIP (Radford et al., 2021), and the multi-scale transformer structure, *i.e.,* Swin-Transformer (Liu et al., 2021), which have both received increasing attention in the area of VP (Bahng et al., 2022a). Our experi-

Table 5: Performance comparison of VP and the proposed AP on CLIP and Swin-Transformer model with different datasets. CLIP with ViT-B/32 and Swin-B with 12 Swin-Transformer blocks pretrained on ImageNet are tested. Other settings follows Tab. 1.

| Dataset | OxfordPets | DTD | EuroSAT | Flowers102 | UCF101 | Food101 | Waterbirds |
|---|---|---|---|---|---|---|---|
| CLIP | | | | | | | |
| VP | 81.97 | 64.43 | 95.54 | 83.74 | 70.42 | 79.61 | 72.42 |
| AP (Ours) | 83.82 | 69.42 | 96.43 | 85.52 | 76.42 | 82.43 | 79.32 |
| Swin-Transformer | | | | | | | |
| VP | 80.42 | 65.39 | 97.23 | 84.48 | 74.41 | 75.72 | 75.22 |
| AP (Ours) | 82.29 | 69.13 | 96.45 | 84.98 | 75.92 | 81.38 | 78.99 |

ments demonstrate that the proposed idea of AP works well even on steering a pretrained CLIP model and Swin-Transformer without changing its parameters. In Fig. 7 and Tab. 5, we demonstrate that our main conclusions about AP still holds for these two architectures well on various datasets. Specifically, in Fig. 7, we show that the layer effect of AP still exists. As both CLIP and Swin-Transformer uses a ViT as its backbone, the observed layer effect mimics that of a ViT-Large/16 as observed before. Specifically, AP prefers to be installed on shallow layers to deep ones in order to obtain the best performance. In Tab. 5, we demonstrate that in various datasets, AP can significantly outperform VP by $1\% \sim 6\%$. These experiments demonstrate the applicability of AP on various model types.

**Ablation studies and additional experiments.** We provide abundant additional experiment results in Appx. B in order to provide discussions on the design of AP and also a comprehensive performance comparison with other methods. In particular, we justified the layer effects more (dataset, model architecture) combinations in Fig. A1 similar to Fig. 4. Besides, we also studied various variants of AP, including AP with different prompt types in Tab. A3, and AP installed in

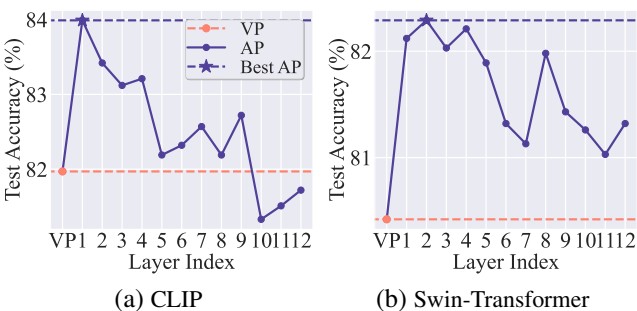

(a) CLIP      (b) Swin-Transformer

Figure 7: The layer effect of AP applied to a (a) CLIP model and (b) Swin-Transformer on the OxfordPets dataset.

multiple layers in Tab. A4. A detailed comparison between AP and other PEFT methods in various experimental settings is also provided, including VPT (Jia et al., 2022) (Tab. A2, Tab. A6, and Fig. A2), LoRA (Hu et al., 2021) (Tab. A8), and SST (Lian et al., 2022) (Tab. A7).

**Limitations and discussions.** We acknowledge a potential limitation of AP lies in its implicit reliance on the size of the pretrained model as a factor for achieving superior accuracy. For compact models like ResNet-18 and ViT-Tiny, while AP enhances the performance of VP, it does not outperform NORM-TUNE. This observation suggests that AP may primarily utilize downstream data to guide or "direct" the existing learned knowledge obtained during pretraining, rather than actively acquiring new knowledge. However, we believe that this limitation does not prevent AP from future applications to larger foundational vision models. We also note that, unlike VP, AP cannot be applied in black-box settings where parameters are inaccessible. However, the primary motivation of this work is to explore the conditions under which VP is effective or ineffective, using AP as an analytical tool to study layer selection preferences for prompting. By doing so, AP broadens the scope of VP, providing deeper insights into its underlying mechanisms under different model settings.

# 7 CONCLUSION

In this paper, we delve into AP (activation prompt) as a means to enhance the conventional input-level VP. We unveil that extending VP to AP yields improved empirical performance and establishes a connection with normalization tuning. Additionally, we investigate the layer preference of AP on CNNs and ViTs both empirically and theoretically. Our experiments demonstrate the superiority of AP over VP, highlighting its efficiency advantages, and showcasing comparable performance to the staet-of-the-art PEFT methods.

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

# APPENDIX

## A EXPERIMENT SETTING DETAILS

**Datasets.** We consider 29 downstream image classification tasks in the target domain across various domains. We show each dataset's attributes in Tab. A1.

| Dataset | \|Train Size\| | Test Size\| | Class Number\| | Batch Size\| | Reference |
|---|---|---|---|---|---|
| **Full-Data Setting** | | | | | |
| Flowers102 | 4093 | 2463 | 102 | 128 | (Nilsback and Zisserman, 2008) |
| DTD | 2820 | 1692 | 47 | 128 | (Cimpoi et al., 2014) |
| UCF101 | 7639 | 3783 | 101 | 128 | (Soomro et al., 2012) |
| Food101 | 50500 | 30300 | 101 | 128 | (Bossard et al., 2014) |
| SVHN | 73257 | 26032 | 10 | 128 | (Netzer et al., 2011) |
| GTSRB | 39209 | 12630 | 43 | 128 | (Houben et al., 2013) |
| EuroSAT | 13500 | 8100 | 10 | 128 | (Helber et al., 2019) |
| OxfordPets | 2944 | 3669 | 37 | 128 | (Parkhi et al., 2012) |
| StanfordCars | 6509 | 8041 | 196 | 128 | (Krause et al., 2013) |
| SUN397 | 15888 | 19850 | 397 | 128 | (Xiao et al., 2010) |
| CIFAR10 | 50000 | 10000 | 10 | 128 | (Krizhevsky et al., 2009) |
| CIFAR100 | 50000 | 10000 | 100 | 128 | (Krizhevsky et al., 2009) |
| CUB-200-2011 | 5394 | 5794 | 200 | 128 | (Wah et al., 2011) |
| NA-Birds | 21536 | 24633 | 55 | 128 | (Van Horn et al., 2015) |
| StanfordDog | 10800 | 8580 | 120 | 128 | (Khosla et al., 2011) |
| OxfordFlowers | 1020 | 6149 | 102 | 128 | (Nilsback and Zisserman, 2008) |
| Waterbirds | 4795 | 5794 | 2 | 128 | (Sagawa et al., 2019) |
| Caltech101 | 4128 | 2465 | 102 | 128 | (Li et al., 2006) |
| Camelyon | 262144 | 32768 | 2 | 128 | (Veeling et al., 2018) |
| **Few-Shot Setting (VTab-1k)** | | | | | |
| CIFAR-100 | 1000 | 10000 | 100 | 128 | (Krizhevsky et al., 2009) |
| Caltech101 | 1000 | 6084 | 102 | 128 | (Li et al., 2006) |
| DTD | 1000 | 47 | 1880 | 128 | (Cimpoi et al., 2014) |
| Flowers102 | 1000 | 6149 | 102 | 128 | (Nilsback and Zisserman, 2008) |
| OxfordPets | 1000 | 3669 | 37 | 128 | (Parkhi et al., 2012) |
| SVHN | 1000 | 26032 | 10 | 128 | (Netzer et al., 2011) |
| Sun397 | 1000 | 21750 | 397 | 128 | (Xiao et al., 2010) |
| Patch Camelyon | 1000 | 32768 | 2 | 128 | (Veeling et al., 2018) |
| EuroSAT | 1000 | 5400 | 10 | 128 | (Helber et al., 2019) |
| Resisc45 | 1000 | 6300 | 45 | 128 | (Cheng et al., 2017) |
| Retinopathy | 1000 | 42670 | 5 | 128 | (Kaggle and EyePacs, 2015) |
| Clevr/count | 1000 | 15000 | 8 | 128 | (Johnson et al., 2017) |
| Clevr/distance | 1000 | 15000 | 6 | 128 | (Johnson et al., 2017) |
| DMLab | 1000 | 22735 | 6 | 128 | (Beattie et al., 2016) |
| KITTI/distance | 1000 | 711 | 4 | 128 | (Geiger et al., 2013) |
| dSprites/location | 1000 | 73728 | 16 | 128 | (Matthey et al., 2017) |
| dSprites/orientation | 1000 | 73728 | 16 | 128 | (Matthey et al., 2017) |
| SmallNORB/azimuth | 1000 | 12150 | 18 | 128 | (LeCun et al., 2004) |
| SmallNORB/elevation | 1000 | 12150 | 9 | 128 | (LeCun et al., 2004) |

Table A1: Dataset attributes and training configs through 29 target image-classification datasets.

**Implementation details.** As we stated in the main manuscript, we, by default, install AP to the input of the thrid-to-last ResNet block and the third Transformer block in ViT-Large/16. For LoRA (Hu et al., 2021), we use the rank $r = 10$ by default. For VPT (Jia et al., 2022), we use a prompt length of 10. We train all the methods for 1000 epochs using an Adam optimizer. For AP, we adopt a learning rate of 0.001 for ResNet family and 0.01 for ViT family without weight decay. For baselines, we adopt the learning rate suggested in the papers or official code repositories. In order to align with the settings of the most parameter efficient fine-tuning methods, for all the prompting-based methods we also tune the classification head as LINEAR-PROBE throughout this work.

## B ADDITIONAL EXPERIMENT RESULTS

**Layer effect study on more datasets.** In Fig. A1, we demonstrate that the layer effects of AP demonstrated in Sec. 4 is general and apply to multiple datasets.

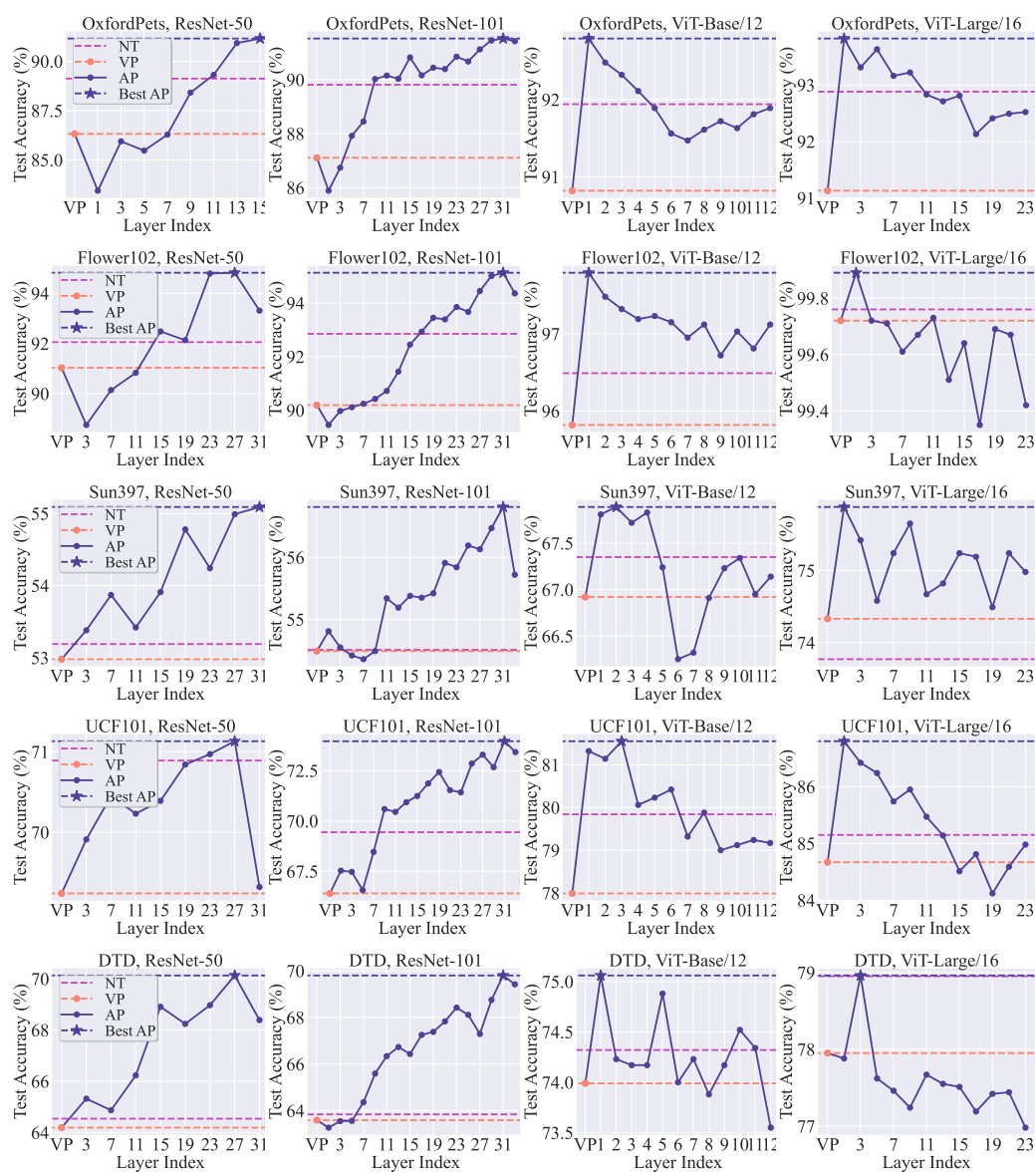

Figure A1: Layer preference of AP with different model architectures on different datasets. CNNs and ViTs exhibit opposite layer preferences.

**Performance of AP in the original experiment setting of VPT.** We conduct an ablation study to strictly follow the experiment settings of VPT, with these results included in Tab. A2. The performance of VPT is directly sourced from Tab. 1 of (Jia et al., 2022). As we can see, the performance as well as efficiency of AP positions itself between VPT-Shallow and VPT-Deep, with an average of 3% performance gain over VPT-Shallow and an average of 3.5% drop compared to VPT-Deep. Regarding these results, we would like to mention that the results of VPT reported in Table 1 of (Jia et al., 2022) are selected based on its best prompt length per dataset, while AP sticks to the same hyper-parameters across all the datasets.

Table A2: Performance comparison of AP with other methods in the setting of VPT (Jia et al., 2022). Specifically, ViT-B/16 pretrained on supervised ImageNet-21k is adopted as the pretrained model. The numbers except AP are directly sourced from VPT (Jia et al., 2022).

| ViT-B/16 (85.8M) | Total Params | FGCV | VTAB-1k Natural | Specialized | Structured |
|---|---|---|---|---|---|
| FULL-FINETUNE | 24.02× | 88.54 | 75.88 | 83.36 | 47.64 |
| LINEAR-PROBE | 1.02× | 79.32 | 68.93 | 77.16 | 26.84 |
| VPT-SHALLOW | 1.04× | 84.62 | 76.81 | 74.66 | 46.98 |
| VPT-DEEP | 1.18× | 89.11 | 78.48 | 82.43 | 54.98 |
| AP (Ours) | 1.11× | 87.33 | 76.59 | 79.32 | 49.98 |

**Ablation study on additional prompt types in AP.** We conduct additional experiments, with the findings presented in Tab. A3. We observed that the originally proposed AP outperforms its new prompt variants studied in Tab. A3 (AP-Product and AP-Concate). We speculate that the advantage of the originally proposed AP may stem from its intrinsic connection to NORM-TUNE, as discussed in the concluding part of Sec. 3.

Table A3: Ablation study on AP with more prompt types. Specifically, instead of using additive prompt in the intermediate layer, AP-PRODUCT uses feature-wise product and AP-CONCATE adopts concatenating prompt.

| | Accuracy | | | Efficiency | | | |
|---|---|---|---|---|---|---|---|
| | **Full-Data** | | | **Train-Time Efficiency** | | | |
| | FGVC | VTAB | Others | Param. # | Memory | Time | Throughput |
| Number of tasks | 5 | 9 | 5 | - | - | - | - |
| FULL-FINETUNE | 91.43 | 91.97 | 93.91 | 304.33 | 41.5 | 520 | 79.58 |
| LINEAR-PROBE | 82.23 | 78.90 | 87.81 | 0.01 | 9.7 | 121 | 79.64 |
| BIAS | 85.32 | 89.84 | 90.41 | 0.29 | 32.9 | 297 | **79.48** |
| LoRA | 86.87 | 89.81 | 91.45 | 1.00 | 33.1 | 363 | **79.43** |
| VPT | 86.05 | 89.97 | 90.64 | 1.24 | 38.6 | 397 | 72.84 |
| ADAPTER | 87.06 | 89.44 | 91.21 | 2.07 | 32.4 | 357 | 63.39 |
| ADAPTERFORMER | **89.18** | **90.69** | **92.08** | 0.65 | 32.3 | 289 | 23.69 |
| AP-PRODUCT | 84.20 | 85.36 | 90.15 | **0.16** | 31.6 | 262 | 79.43 |
| AP-CONCATE | 83.29 | 82.42 | 89.13 | **0.12** | 31.4 | 261 | 79.47 |
| AP | 85.30 | 90.25 | 91.09 | **0.16** | 31.6 | 262 | 79.43 |

**Application of AP to multiple layers.** We implement AP with multiple layers, and we show the results in Tab. A4. Our findings indicate that the layer addition of AP does not yield significant improvements in performance. This observation is significant as it suggests that applying AP to a single, carefully selected layer can achieve comparable performance to more extensive applications. This underscores the efficiency of AP, affirming its value in settings where computational resources are a concern.

Table A4: Ablation study on the number of layers installed with AP. In particular, for AP-3 and AP-5, AP are installed on the input of the first 3 and 5 blocks of the pretrained ViT-L. Other experiment settings follow Tab. 1, and Tab. 2.

| | Accuracy | | | Efficiency | | | |
|---|---|---|---|---|---|---|---|
| | **Full-Data** | | | **Train-Time Efficiency** | | | |
| | FGVC | VTAB | Others | Param. # | Memory | Time | Throughput |
| Number of tasks | 5 | 9 | 5 | - | - | - | - |
| FULL-FINETUNE | 91.43 | 91.97 | 93.91 | 304.33 | 41.5 | 520 | 79.58 |
| LINEAR-PROBE | 82.23 | 78.90 | 87.81 | 0.01 | 9.7 | 121 | 79.64 |
| BIAS | 85.32 | 89.84 | 90.41 | 0.29 | 32.9 | 297 | 79.48 |
| LoRA | 86.87 | 89.81 | 91.45 | 1.00 | 33.1 | 363 | 79.43 |
| VPT | 86.05 | 89.97 | 90.64 | 1.24 | 38.6 | 397 | 72.84 |
| ADAPTER | 87.06 | 89.44 | 91.21 | 2.17 | 32.4 | 357 | 63.39 |
| ADAPTERFORMER | **89.18** | **90.69** | **92.08** | 0.65 | 32.3 | 289 | 23.69 |
| AP-3 | 85.41 | 90.38 | 91.21 | 0.46 | 47.8 | 297 | 79.43 |
| AP-5 | 85.49 | 90.49 | 91.31 | 0.76 | 69.7 | 348 | 79.43 |
| AP | 85.30 | 90.25 | 91.09 | 0.16 | 31.6 | 262 | 79.43 |

**Performance comparison with re-initialized classification head.** We carried out an ablation experiment using re-initialized classification head. This will influence the tunable parameter counts of LINEAR-PROBE and other methods involved. As we can see, the results in Tab. A5 are nearly identical to our previous findings in Tab. 4 that AP shows a competitive performance and efficiency compared with other strong PEFT baselines.

Table A5: Performance comparison between AP and SOTA PEFT methods on ViT-Large/16 with re-initialized classification head. Experiment settings follow Tab. 1, and Tab. 2.

| | Accuracy | | | Efficiency | | | |
| | **Full-Data** | | | **Train-Time Efficiency** | | | |
| | FGVC | VTAB | Others | Param. # | Memory | Time | Throughput |
| Number of tasks | 5 | 9 | 5 | - | - | - | - |
| FULL-FINETUNE | 91.43 | 91.97 | 93.91 | 304.33 | 41.5 | 520 | 79.58 |
| LINEAR-PROBE | 82.31 | 78.43 | 87.71 | 0.01 | 8.1 | 121 | 79.69 |
| BIAS | 85.49 | 89.47 | 90.85 | 0.29 | **27.4** | 297 | **79.51** |
| LoRA | 86.49 | 89.74 | 91.49 | 1.00 | 32.5 | 363 | 71.47 |
| VPT | 86.15 | 90.13 | 90.88 | 1.24 | 37.2 | 397 | 72.91 |
| ADAPTER | 87.14 | 89.12 | 91.01 | 2.07 | 31.1 | 357 | 63.78 |
| ADAPTERFORMER | 89.24 | 90.49 | 92.21 | 0.65 | 31.1 | 289 | 23.82 |
| AP | 85.32 | 90.12 | 91.11 | **0.16** | 30.2 | **262** | 79.54 |

**Comparison to VPT with other prompt lengths.** We conducted an experiment to implement VPT-Deep using a smaller prompt token length 10 (VPT-10). The results, presented in Tab. A6, indicate that VPT-10's performance is comparable to VPT-50 in Tab. 4, albeit with enhanced efficiency.

Table A6: Performance comparison between AP and VPT with different prompt lengths on ViT-Large/16. Experiment settings follow Tab. 1, and Tab. 4.

| | Accuracy | | | Efficiency | | | |
| | **Full-Data** | | | **Train-Time Efficiency** | | | |
| | FGVC | VTAB | Others | Param. # | Memory | Time | Throughput |
| Number of tasks | 5 | 9 | 5 | - | - | - | - |
| FULL-FINETUNE | 91.43 | 91.97 | 93.91 | 304.33 | 41.5 | 520 | 79.58 |
| LINEAR-PROBE | 82.23 | 78.90 | 87.81 | 0.01 | 9.7 | 121 | 79.64 |
| VPT-10 | 86.34 | 89.24 | 90.14 | 0.25 | 33.7 | 334 | 76.35 |
| VPT-50 | 86.05 | 89.97 | 90.64 | 1.24 | 38.6 | 397 | 72.84 |
| AP | 85.30 | 90.25 | 91.09 | **0.16** | **31.6** | **262** | **79.43** |

**Layerwise comparison between AP and VPT-Deep.** We conduct an experiment for a more detailed layer-wise evaluation in Fig. A2. These additional results highlight a consistent layer-architecture influence on VPT-Deep, akin to what we initially observed in our original AP design. This outcome is not unexpected, considering that the implementation of VPT-Deep essentially converges with that of AP when a specific network layer is selected for prompting. The key divergence lies in the prompt design approach: VPT-Deep favors concatenation, whereas AP opts for addition in prompt design. It is worth noting that, in the context of single-layer prompting, the efficacy of concatenation in prompt design is comparatively lower than that of addition.

**Comparison with additional PEFT methods.** We conduct an experiment and report the results of SSF in Tab. A7. In particular, we can see SSF is also a competitive method among all the baselines but is still under AdapterFormer. Compared to AP, SSF yields better performance for the FGVC benchmark but leads to slightly worse accuracy for the VTAB benchmark. In general, SSF ranks approximately the second or the third place among all the PEFT methods.

**Comparison with LoRA of different rank values.** We conduct additional experiments on the hyper-parameters of LoRA, namely the rank $r$. In Tab. 4, the rank $r$ is adopted to 10 by default. In Tab. A8, we explore more rank values varying from 1 to 50. We can see that the performance of LoRA increases with the larger rank values, but the difference between $r = 10$ and $r = 50$ is insignificant. In contrast, the efficiency of LoRA will drop significantly with a rank larger than 10. In

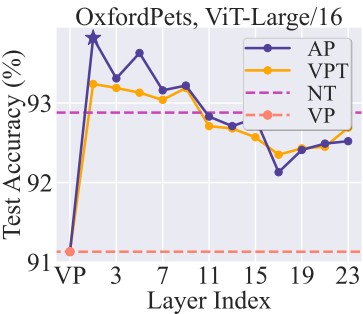

Figure A2: Layer-wise performance comparison between AP and VPT on OxfordPets.

Table A7: Performance comparison of AP with more PEFT methods (SSF (Lian et al., 2022)). Experiment settings follow Tab. 1 and Tab. 4.

| | Accuracy | | | Efficiency | | | |
|---|---|---|---|---|---|---|---|
| | **Full-Data** | | | | **Train-Time Efficiency** | | |
| | FGVC | VTAB | Others | Param. # | Memory | Time | Throughput |
| Number of tasks | 5 | 9 | 5 | - | - | - | - |
| FULL-FINETUNE | 91.43 | 91.97 | 93.91 | 304.33 | 41.5 | 520 | 79.58 |
| LINEAR-PROBE | 82.23 | 78.90 | 87.81 | 0.01 | 9.7 | 121 | 79.64 |
| BIAS | 85.32 | 89.84 | 90.41 | 0.29 | 32.9 | 297 | **79.48** |
| LoRA | 86.87 | 89.81 | 91.45 | 1.00 | 33.1 | 363 | **79.43** |
| VPT | 86.05 | 89.97 | 90.64 | 1.24 | 38.6 | 397 | 72.84 |
| ADAPTER | 87.06 | 89.44 | 91.21 | 2.17 | 32.4 | 357 | 63.39 |
| ADAPTERFORMER | **89.18** | **90.69** | **92.08** | 0.65 | 32.3 | 289 | 23.69 |
| SSF | 87.32 | 89.43 | 92.21 | 0.48 | 34.7 | 299 | **79.49** |
| AP | 85.30 | 90.25 | 91.09 | **0.16** | **31.6** | **262** | **79.43** |

order to strike a balance between performance and efficiency, we adopt the rank value of 10 as the default value in this work.

**Ablation study on the influence of different data sizes.** We recognize that data size significantly influences performance. To ensure that our conclusions generalize well, we conducted an ablation study on FULL-FINETUNE, VP, and AP, varying the training data ratio from 10% to 100% on datasets with large training sizes (Camelyon, FOOD101, CIFAR10). The results are shown in Figure A3. Results show that FULL-FINETUNE benefits the most from larger datasets. However, AP consistently outperforms VP, regardless of data size, reinforcing that AP is a better design than VP for both few- and many-shot settings.

Table A8: Ablation study on performance of LoRA with different rank values. Experiment settings follow Tab. 1 and Tab. 4.

| | Accuracy | | | Efficiency | | | |
| | Full-Data | | | | Train-Time Efficiency | | |
| | FGVC | VTAB | Others | Param. # | Memory | Time | Throughput |
|---|---|---|---|---|---|---|---|
| Number of tasks | 5 | 9 | 5 | - | - | - | - |
| FULL-FINETUNE | 91.43 | 91.97 | 93.91 | 304.33 | 41.5 | 520 | 79.58 |
| LINEAR-PROBE | 82.23 | 78.90 | 87.81 | 0.01 | 9.7 | 121 | 79.64 |
| LoRA-1 | 84.43 | 88.21 | 90.07 | 0.04 | 10.43 | 139 | 79.43 |
| LoRA-10 | 86.87 | 89.81 | 91.45 | 1.00 | 33.1 | 363 | 79.43 |
| LoRA-20 | 86.93 | 90.23 | 91.35 | 4.38 | 33.1 | 443 | 79.43 |
| LoRA-50 | 87.23 | 90.41 | 91.97 | 12.22 | 57.2 | 589 | 79.43 |
| AP | 85.30 | 90.25 | 91.09 | 0.16 | 31.6 | 262 | 79.43 |

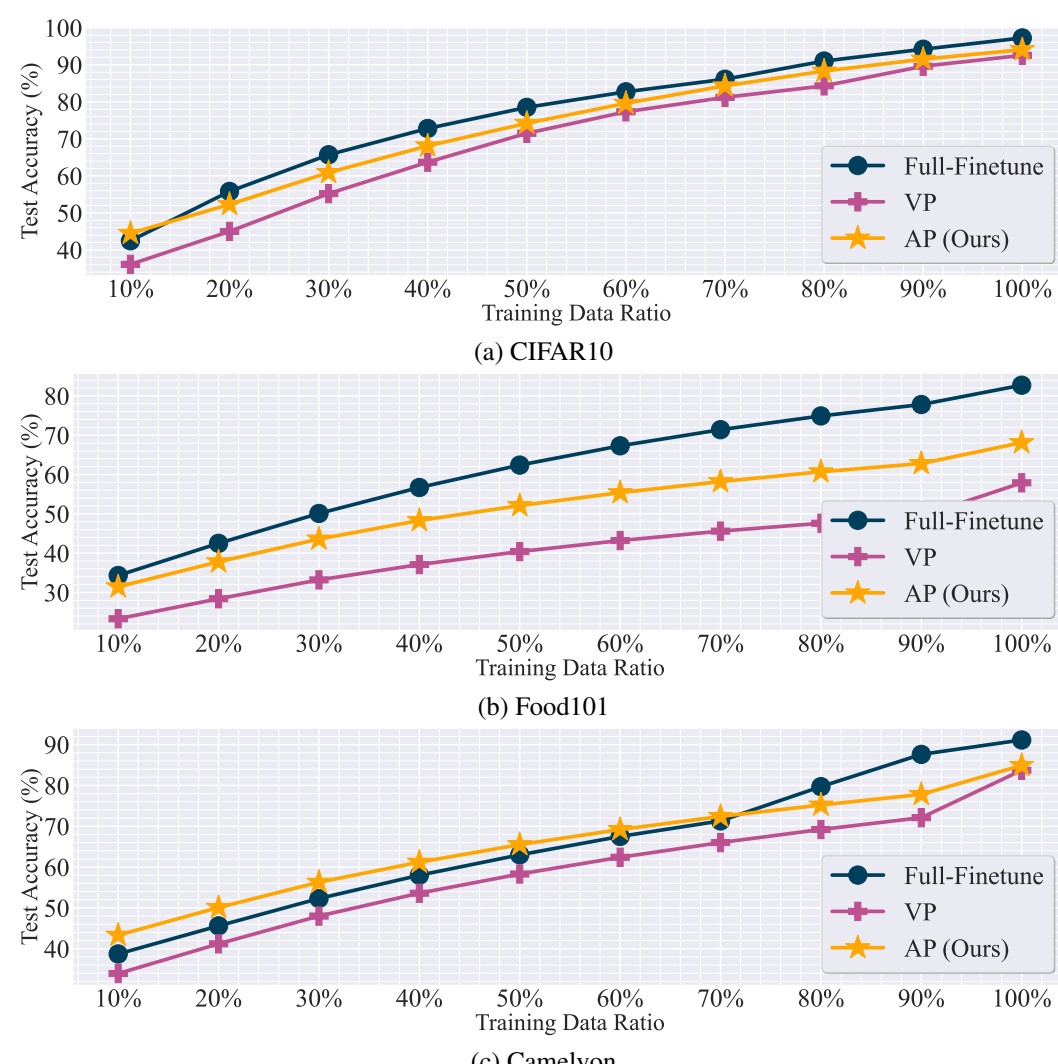

(a) CIFAR10

(b) Food101

(c) Camelyon

Figure A3: Performance of ResNet101 trained with varying sizes of available training data on (a) CIFAR10, (b) Food101, and (c) Camelyon. All other experimental settings strictly follow those in Tab. 1.

## C THEORETICAL DETAILS

### C.1 MODEL ARCHITECTURE

We define the general definition of the model architecture CNN, ViT in this section.

**CNN**: We follow the architecture of ResNet (), which stacks multiple residual blocks plus an input and an output layer. Each residual block includes several convolutional layers and a skip connection. For the input $z_{\text{in}}^{(l)}$ to the $l$-th convolutional layer, where $l \in [L]$, the output $z_{\text{out}}^{(l)}$ can be computed as

$$z^{(l)} = \text{Conv}(z_{\text{in}}^{(l)}; W_1^{(l)}), \ z_{\text{out}}^{(l)} = \text{relu}(\text{BN}(z^{(l)})) \tag{A1}$$

where $z_{\text{in}}^{(0)} = x$. Conv($\cdot$) and BN denote the Convolution operation and the Batch Normalization, respectively. The output $\hat{y} = \text{FC}(\text{Pooling}(z_{\text{out}}^{(L)}))$, where FC($\cdot$) denotes fully-connected layer.

**ViT**: The architecture of Vision Transformer is defined in (). For the input $z_{\text{in}}^{(l)}$ to the $l$-th Transformer layer, we first let $z^{(l)} = z_{\text{in}}^{(l)}$. Then, the output $z_{\text{out}}^{(l)}$ can be computed as

$$z^{(l)} = \text{MSA}(\text{LN}(z^{(l)})) + z^{(l)}, \ z_{\text{out}}^{(l)} = \text{MLP}(\text{LN}(z^{(l)})) + z^{(l)}, \tag{A2}$$

where $z_{\text{in}}^{(0)} = x$. MSA($\cdot$) and LN($\cdot$) denote the Multi-Head Self-attention and Layer Normalization, respectively. For an $L$-layer ViT, the output $\hat{y} = \text{Out}(H_{\text{out}}^{(L)})$, where Out($\cdot$) denotes the output layer.

### C.2 PROPOSITION 1 AND ITS PROOF

We first provide a full definition of NORM-TUNE.

NORM-TUNE is a method where only the Batch Normalization layers for CNNs or Layer Normalization for ViTs are trainable. Consider a batch of the $l$-th-layer features $z_1^{(l)}, z_2^{(l)}, \cdots, z_B^{(l)}$ defined in (A1) and (A2), where $z_b^{(l)} = [z_{b,\cdot,1}^{(l)}, z_{b,\cdot,2}^{(l)}, \cdots, z_{b,\cdot,P'}^{(l)}] = \in \mathbb{R}^{D' \times P'}$, $z_{b,\cdot,p}^{(l)} \in \mathbb{R}^{D'}$ for $b \in [B]$ and $p \in [P']$. $B$ is the batch size, $D'$ denotes the number of channels or token dimension, and $P'$ denotes the size of the feature map or token length. We can formulate the Normalization on $h_{b,d,p}^{(l)}$, the $d$-th dimension of $h_{b,\cdot,p}^{(l)}$, as follows.

$$\textbf{BN}: \mu_d = \sum_{b=1}^{B}\sum_{p=1}^{P'} \frac{z_{b,d,p}^{(l)}}{BP'}, \ \sigma_d^2 = \sum_{b=1}^{B}\sum_{p=1}^{P'} \frac{(z_{b,d,p}^{(l)} - \mu_d)^2}{BP'}, \ \text{BN}(z_{b,d,p}^{(l)}) = \gamma_d \frac{z_{b,d,p}^{(l)} - \mu_d}{\sigma_d} + \beta_d,$$

$$\textbf{LN}: \mu_{b,p} = \sum_{d=1}^{D'} \frac{z_{b,d,p}^{(l)}}{D'}, \ \sigma_{b,p}^2 = \sum_{d=1}^{D'} \frac{(z_{b,d,p}^{(l)} - \mu_{b,p})^2}{D'}, \ \text{LN}(z_{b,d,p}^{(l)}) = \gamma_d \frac{z_{b,d,p}^{(l)} - \mu_{b,p}}{\sigma_{b,p}} + \beta_d,$$

$$\tag{A3}$$

where $\gamma_d$, $\beta_d$ are trainable parameters for $d \in [D']$. Then, we present a full statement of Proposition 1.

**Proposition 1** *Without the assumption that the input to the batch (or layer) normalization layer has zero mean and unit variance for each dimension (or token), we have the following conclusion:*

AP *on the $l$-th layer is the same as* NORM-TUNE *on the $l$-th layer, if*

- *for CNNs, $\gamma_d/\sigma_d = 1$, and all $\delta_p$'s added to $z_b^{(l)}$ are the same as $\delta$, $\beta_d = w_d^{(l)}\delta_* + \mu_d$ for all $d \in [D']$, where $\delta_* = \delta_i^{(l)}$ for $i \in [P']$;*

- *for ViTs, $\gamma_d/\sigma_{b,p} = 1$, and $\mu_{b,p}$'s are the same as $\mu_p$, $p \in [P']$ among all $b \in [B]$ for ViTs, $\beta_d = \delta_{p,d}^{(l)} + \mu_p$ for all $d \in [D']$, $p \in [P']$.*

**Proof:**

For BN, note that

$$\mathrm{BN}(z_{b,d,p}^{(l)}) = \gamma_d \frac{z_{b,d,p}^{(l)} - \mu_d}{\sigma_d} + \beta_d = \frac{\gamma_d}{\sigma_d} z_{b,d,p}^{(l)} + \beta_d - \frac{\mu_d \gamma_d}{\sigma_d} \tag{A4}$$

where

$$z_{b,d,p}^{(l)} = \boldsymbol{w}_d^{(l)} \boldsymbol{z}_{b,\cdot,p}^{(l-1)}, \ \boldsymbol{z}_{b,\cdot,p}^{(l-1)} = \boldsymbol{x}_{b,\cdot,p} \tag{A5}$$

When adding the prompt $\boldsymbol{\delta}_p^{(l)}$, we have the output

$$\boldsymbol{w}_d^{(l)}(\boldsymbol{z}_{b,\cdot,p}^{(l-1)} + \boldsymbol{\delta}_p^{(l)}) \tag{A6}$$

We then need the equation

$$\frac{\gamma_d}{\sigma_d} z_{b,d,p}^{(l)} + \beta_d - \frac{\mu_d \gamma_d}{\sigma_d} = \boldsymbol{w}_d^{(l)}(\boldsymbol{z}_{b,\cdot,p}^{(l-1)} + \boldsymbol{\delta}_p^{(l)}) \tag{A7}$$

Given $\gamma_d/\sigma_d = 1$, we have

$$\beta_d = \boldsymbol{w}_d^{(l)} \boldsymbol{\delta}_p^{(l)} + \mu_d \tag{A8}$$

Suppose that $\mu_d = 0$ for $d \in [D']$ and $\boldsymbol{\delta}_p^{(l)} = \boldsymbol{\delta}_*$ for $p \in [P']$, we can obtain

$$\beta_d = \boldsymbol{w}_d^{(l)} \boldsymbol{\delta}_* \tag{A9}$$

For LN, we need

$$\mathrm{LN}(z_{b,d,p}^{(l)}) = \gamma_d \frac{z_{b,d,p}^{(l)} - \mu_{b,p}}{\sigma_{b,p}} + \beta_d = \frac{\gamma_d}{\sigma_{b,p}} z_{b,d,p}^{(l)} + \beta_d - \frac{\gamma_d \mu_{b,p}}{\sigma_{b,p}} = z_{b,d,p}^{(l)} + \delta_{p,d}^{(l)} \tag{A10}$$

Given $\gamma_d/\sigma_{b,p} = 1$ and $\boldsymbol{\mu}_{b,p} = \boldsymbol{\mu}_p$ for $b \in [B]$, we have

$$\beta_d = \delta_{p,d}^{(l)} + \mu_p \tag{A11}$$

Suppose that $\mu_p = 0$, $p \in [P']$ and let $\boldsymbol{\delta}_p^{(l)} = \boldsymbol{\delta}_*$, $p \in [P']$, we can obtain

$$\boldsymbol{\beta} = \boldsymbol{\delta}_* \tag{A12}$$

### C.3 PROOF OF LEMMA 1

Before we provide the proof, we state the formulation of a single-head and two-layer ViT, the full assumption on the data model, and the pretrained model in detail.

Let $\boldsymbol{x}_{n(\cdot,j)}$ be the $j$-th patch/token of $\boldsymbol{x}_n$, $j \in [P]$. The corresponding 1-st-layer output is $\boldsymbol{z}_{n(\cdot,j)}$. Denote the $j$-th patch/token of $\boldsymbol{x}_n$ or $\boldsymbol{z}_n$ after introducing the AP, $\boldsymbol{\delta}^{(h)}$, as $\boldsymbol{x}_n[\boldsymbol{\delta}_j^{(h)}]$ and $\boldsymbol{z}_n[\boldsymbol{\delta}_j^{(h)}] = (\boldsymbol{z}_n[\boldsymbol{\delta}_1^{(h)}], \cdots, \boldsymbol{z}_n[\boldsymbol{\delta}_P^{(h)}])$, respectively.

Following (Dosovitskiy et al., 2020), we consider a single-head self-attention parameterized by $\boldsymbol{W}_Q^{(l)}$, $\boldsymbol{W}_K^{(l)}$, and $\boldsymbol{W}_V^{(l)}$ in the $l$-th layer. The shapes of these matrices are $m$ by $d$ if $l = 1$ and $m$ by $m$ if $l = 2$. Denote $\boldsymbol{W}^{(l)} = \boldsymbol{W}_K^{(l)\top} \boldsymbol{W}_Q^{(l)}$, $l = 1, 2$. The MLP layer is a two-layer perceptron with $m \times m$-dimensional parameters $\boldsymbol{W}_O^{(l)}$, $\boldsymbol{W}_U^{(l)}$, and Relu activation. The output layer is a fully-connected layer with $\boldsymbol{a}_1, \cdots, \boldsymbol{a}_P$ where $\boldsymbol{a}_l \in \mathbb{R}^m$. Then, a two-layer ViT can be written as

$$f_{\boldsymbol{\theta}}(\boldsymbol{x}_n, \boldsymbol{\delta}^{(h)}) = \sum_{k=1}^P \boldsymbol{a}_k^\top \boldsymbol{W}_U^{(2)} \mathrm{Relu}(\boldsymbol{W}_O^{(2)} \boldsymbol{W}_V^{(2)} \boldsymbol{z}_n[\boldsymbol{\delta}^{(h)}] \mathrm{softmax}(\boldsymbol{z}_n[\boldsymbol{\delta}^{(h)}]^\top \boldsymbol{W}^{(2)} \boldsymbol{z}_n[\boldsymbol{\delta}_k^{(h)}])),$$

$$\boldsymbol{z}_n[\boldsymbol{\delta}_k^{(h)}] = \boldsymbol{W}_U^{(1)} \mathrm{Relu}(\sum_{s=1}^P \boldsymbol{W}_O^{(1)} \boldsymbol{W}_V^{(1)} \boldsymbol{x}_n[\boldsymbol{\delta}_s^{(h)}] \mathrm{softmax}(\boldsymbol{x}_n[\boldsymbol{\delta}_s^{(h)}]^\top \boldsymbol{W}^{(1)} \boldsymbol{x}_n[\boldsymbol{\delta}_k^{(h)}])), \tag{A13}$$

The AP is restated as

$$\begin{cases} \boldsymbol{x}_n[\boldsymbol{\delta}_j^{(h)}] = \boldsymbol{x}_{n(\cdot,j)} + \boldsymbol{\delta}_j^{(h)}, \boldsymbol{z}_n[\boldsymbol{\delta}_j^{(h)}] \text{ as defined in (A13)}, & \text{if } h = 1, \\ \boldsymbol{x}_n[\boldsymbol{\delta}_j^{(h)}] = \boldsymbol{x}_{n(\cdot,j)}, \boldsymbol{z}_n[\boldsymbol{\delta}_j^{(h)}] = \boldsymbol{z}_{n(\cdot,j)} + \boldsymbol{\delta}_j^{(h)}, & \text{if } h = 2, \end{cases} \tag{A14}$$

We use Hinge loss $\ell(\boldsymbol{x}_n.y_n) = \max\{0, 1/P - y_n f_{\boldsymbol{\theta}}(\boldsymbol{x}_n, \boldsymbol{\delta}^{(h)})\}$ as the loss function.

**Data model** The patch/token $\boldsymbol{x}_{n(\cdot,j)}$ is a noisy version of patterns, i.e., $\boldsymbol{x}_{n(\cdot,j)} = \boldsymbol{v}_l + \epsilon_j^n$, where $\boldsymbol{v}_l$, $l = 1, 2, 3, 4$ is a pattern and $\epsilon_j^n \sim \mathcal{N}(0, \sigma^2)$ is a Gaussian noise, $\sigma \leq O(1/P)$. $\boldsymbol{v}_1, \boldsymbol{v}_2, \boldsymbol{v}_3, \boldsymbol{v}_4$ are all unit norm and orthogonal to each other except the pairs of $\boldsymbol{v}_3$ and $\boldsymbol{v}_4$. $\boldsymbol{v}_3^\top \boldsymbol{v}_4 = \zeta \in (-1, 0)$. In each sample $\boldsymbol{x}_n$, only one patch/token $\boldsymbol{x}_{n(\cdot,j)}$ corresponds to either $\boldsymbol{v}_1$ or $\boldsymbol{v}_2$, while other $P - 1$ patches/tokens correspond to either $\boldsymbol{v}_3$ or $\boldsymbol{v}_4$. $\boldsymbol{v}_1, \boldsymbol{v}_2$ are called discriminative patterns that decide the label. $\boldsymbol{v}_3, \boldsymbol{v}_4$ are non-discriminative patterns that work as the image background. For instance, if one patch is the noisy version of $\boldsymbol{v}_1$ ($\boldsymbol{v}_2$), then $y^n = 1$ ($y^n = -1$).

**Pretrained model** The pretraining stage is assumed to learn a task where all patterns $\{\boldsymbol{v}_1, \boldsymbol{v}_2, \boldsymbol{v}_3, \boldsymbol{v}_4\}$ are key features, where each data contains two types of patterns. The label is determined by the number of $\boldsymbol{v}_1$ or $\boldsymbol{v}_3$ compared with the number of $\boldsymbol{v}_2$ or $\boldsymbol{v}_4$. Inspired by the finding that some trained "lucky" hidden neurons represent discriminative features from existing theoretical works (Li et al., 2023b) on ViTs, we accordingly set the neurons of feed-forward-networks $\boldsymbol{W}_O^{(i)}$ in (A13), $i = 1, 2$ as pattern representations of that layer and ignore "unlucky" neurons, which has a trivial effect on the output. To be more specific, for the 1st layer, we set a $1/4$ fraction of neurons to be $\boldsymbol{v}_i$, $i = 1, 2, 3, 4$, and for the 2nd layer, we set a $1/4$ fraction of neurons to be $\boldsymbol{e}_i$, $i = 1, 2, 3, 4$, i.e., the 2nd-layer pattern representations. $\boldsymbol{W}_U^{(1)} = \boldsymbol{W}_U^{(2)} = \boldsymbol{I}$. $a_{l(i)}$ equal $1/(mP)$ for neurons of $\boldsymbol{e}_1$ and $\boldsymbol{e}_3$, and they equal $-1/(mP)$ for neurons of $\boldsymbol{e}_2$ and $\boldsymbol{e}_4$. For ViTs, we follow the orthogonal embedding assumption in (Oymak et al., 2023; Li et al., 2023b; Zhang et al., 2023b; Li et al., 2023c; Huang et al., 2023b; Li et al., 2023d; 2024a;b;c; Chen et al.) and set $\boldsymbol{W}_Q^{(1)} = \beta_1 \boldsymbol{I}$, $\boldsymbol{W}_K^{(1)} = \beta_1 \boldsymbol{P}_x^{(1)}$, $\boldsymbol{W}_Q^{(2)} = \beta_2 \boldsymbol{I}$, $\boldsymbol{W}_K^{(2)} = \beta_2 \boldsymbol{P}_x^{(2)}$, $\boldsymbol{W}_V^{(1)} = \boldsymbol{P}_x^{(1)}$, $\boldsymbol{W}_V^{(2)} = \boldsymbol{P}_x^{(2)}$ for simplicity, where $\beta_1 = \Theta(1)$, $\beta_2 = \Theta(1)$, $\boldsymbol{I}$ is the identity matrix, and $\boldsymbol{P}_x^{(1)}$ and $\boldsymbol{P}_x^{(2)}$ are permutation matrices.

Then, we present the proof of Lemma 1.

**Proof:**

Without loss of generality, we focus on studying the data where $\boldsymbol{v}_1$ is the discriminative pattern, and $\boldsymbol{v}_4$ is the non-discriminative pattern.

For ViTs, note that the permutation matrix $\boldsymbol{P}_x^{(1)}$ changes the location of the pattern $\boldsymbol{v}_1$ to another place with a distance of at least $d_A$. By computing the feature correlation for the pattern $\boldsymbol{v}_1$, we have

$$\beta_1^2 > 0, \tag{A15}$$

which means the the pattern $\boldsymbol{v}_1$ has the largest correlation with $\boldsymbol{v}_1$. Hence, the pattern of $\boldsymbol{v}_1$ is a global feature. For the feature correlation of the pattern $\boldsymbol{v}_4$, we have

$$\beta_1^2 > 0, \tag{A16}$$

which means the the pattern $\boldsymbol{v}_4$ has the largest correlation with $\boldsymbol{v}_4$. Hence, the pattern of $\boldsymbol{v}_4$ is a global feature because the distance between two $\boldsymbol{v}_4$ patterns is at most 1. Since that there will be one $\boldsymbol{v}_4$ token corresponding to a $\boldsymbol{v}_1$ token after the permutation, there will be a contribution of distance 1 to the average distance. The average attention distance of the first layer is

$$\frac{1}{P} \sum_{i=1}^{P} |i - \arg \max_{j \in [P]} \langle \boldsymbol{k}_j, \boldsymbol{q}_i \rangle| = \frac{1 + d_A}{P} \tag{A17}$$

After the first layer, the feature of the $\boldsymbol{v}_1$ token becomes

$$\frac{e^{\beta_1^2}}{e^{\beta_1^2} + P - 1} \boldsymbol{v}_1 + \frac{P - 1}{e^{\beta_1^2} + P - 1} \boldsymbol{v}_4 := \lambda_1 \boldsymbol{v}_1 + (1 - \lambda_1) \boldsymbol{v}_4, \tag{A18}$$

while the feature of the $\boldsymbol{v}_4$ token becomes

$$\frac{1}{(P - 1)e^{\beta_1^2} + 1} \boldsymbol{v}_1 + \frac{(P - 1)e^{\beta_1^2}}{(P - 1)e^{\beta_1^2} + 1} \boldsymbol{v}_4 := \lambda_2 \boldsymbol{v}_1 + (1 - \lambda_2) \boldsymbol{v}_4, \tag{A19}$$

Here $1/2 > \lambda_1 > \lambda_2 > 0$. Therefore, we have

$$(\lambda_1 \boldsymbol{v}_1 + (1 - \lambda_1) \boldsymbol{v}_4)^\top (\lambda_1 \boldsymbol{v}_1 + (1 - \lambda_1) \boldsymbol{v}_4 - \lambda_2 \boldsymbol{v}_1 - (1 - \lambda_2) \boldsymbol{v}_4)$$
$$= (2\lambda_1 - 1)(\lambda_1 - \lambda_2) < 0 \tag{A20}$$

$$(\lambda_2 \boldsymbol{v}_1 + (1 - \lambda_2)\boldsymbol{v}_4)^\top (\lambda_2 \boldsymbol{v}_1 + (1 - \lambda_2)\boldsymbol{v}_4 - \lambda_1 \boldsymbol{v}_1 - (1 - \lambda_1)\boldsymbol{v}_4)$$
$$= (2\lambda_2 - 1)(\lambda_2 - \lambda_1) > 0 \tag{A21}$$

Therefore, the feature from the token of $\boldsymbol{v}_4$ has the largest correlation with the token of both $\boldsymbol{v}_1$ and $\boldsymbol{v}_4$. Since there exists a $\boldsymbol{v}_4$ token close to $\boldsymbol{v}_1$ token with a distance of at most 1, we have that both $\boldsymbol{v}_1$ and $\boldsymbol{v}_4$ tokens become local features. Then, the average attention distance of the second layer is

$$\frac{1}{P} \sum_{i=1}^{P} |i - \arg\max_{j \in [P]} \langle \boldsymbol{k}_j, \boldsymbol{q}_i \rangle| = \frac{1}{P} \tag{A22}$$

### C.4    PROOF OF THEOREM 1

We first present two lemmas. One can observe that Theorem 1 is a combination of these two lemmas. Therefore, the proof of Theorem 1 is exactly the same as the proof of these two lemmas.

**Lemma 2** *For a two-layer single-head Transformer*

$$f_{\boldsymbol{\theta}}(\boldsymbol{x}_n, \boldsymbol{\delta}) = \sum_{l=1}^{P} \sum_{i=1}^{m} a_{l(i)}^\top Relu(\sum_{j=1}^{P} \boldsymbol{W}_{O_{2(i,\cdot)}} \boldsymbol{W}_{V_2}(\boldsymbol{z}_{n(\cdot,j)} + \boldsymbol{\delta}_j^{(h)})$$
$$\cdot softmax((\boldsymbol{z}_{n(\cdot,j)} + \boldsymbol{\delta}_j^{(h)})^\top \boldsymbol{W}_{K_2}^\top \boldsymbol{W}_{Q_2}(\boldsymbol{z}_{n(\cdot,l)} + \boldsymbol{\delta}_l^{(h)}))) \tag{A23}$$

*where*

$$\boldsymbol{z}_{n(\cdot,j)} = Relu(\sum_{s=1}^{P} \boldsymbol{W}_{O_1} \boldsymbol{W}_{V_1} \boldsymbol{x}_{n(\cdot,s)} softmax(\boldsymbol{x}_{n(\cdot,s)}^\top \boldsymbol{W}_{K_1}^\top \boldsymbol{W}_{Q_1} \boldsymbol{x}_{n(\cdot,j)})) \tag{A24}$$

*as long as the batch size and the required number of iterations satisfy*

$$B \geq \Omega(1), \quad T = \frac{\eta^{-1} P^2 \log P}{(1 - \sigma)^{-1}}, \tag{A25}$$

*where $\sigma \leq \Theta(P^{-1})$, training $\boldsymbol{\delta}^{(h)}$, $h = 2$ with SGD returns a model with zero generalization error.*

**Lemma 3** *For a two-layer single-head Transformer*

$$f_{\boldsymbol{\theta}}(\boldsymbol{x}_n, \boldsymbol{\delta}) = \sum_{l=1}^{P} \sum_{i=1}^{m} a_{l(i)}^\top Relu(\sum_{j=1}^{P} \boldsymbol{W}_{O_{2(i,\cdot)}} \boldsymbol{W}_{V_2} \boldsymbol{z}_{n(\cdot,j)} softmax(\boldsymbol{z}_{n(\cdot,j)}^\top \boldsymbol{W}_{K_2}^\top \boldsymbol{W}_{Q_2} \boldsymbol{z}_{n(\cdot,l)})) \tag{A26}$$

*where*

$$\boldsymbol{z}_{n(\cdot,j)} = Relu(\sum_{s=1}^{P} \boldsymbol{W}_{O_1} \boldsymbol{W}_{V_1}(\boldsymbol{x}_{n(\cdot,s)} + \boldsymbol{\delta}_s^{(h)}) softmax((\boldsymbol{x}_{n(\cdot,s)} + \boldsymbol{\delta}_s^{(h)})^\top \boldsymbol{W}_{K_1}^\top \boldsymbol{W}_{Q_1}(\boldsymbol{x}_{n(\cdot,j)} + \boldsymbol{\delta}_j^{(h)}))) \tag{A27}$$

*as long as the batch size and the required number of iterations satisfy*

$$B \geq \Omega(1), \quad T = \frac{\eta^{-1} P}{(1 - P\sigma)^{-1}(1 + \gamma)}, \tag{A28}$$

*where $\sigma \leq O(P^{-1})$, training $\boldsymbol{\delta}^{(h)}$, $h = 1$ with SGD returns a model with zero generalization error, where $\gamma := \boldsymbol{v}_3^\top \boldsymbol{v}_4 \in (-1, 0)$.*

#### C.4.1    PROOF OF LEMMA 2

**Proof:**

For $h = 2$,

$$f_{\boldsymbol{\theta}}(\boldsymbol{x}_n, \boldsymbol{\delta}^{(h)}) = \sum_{l=1}^{P} \sum_{i=1}^{m} a_{l(i)}^\top Relu(\sum_{s=1}^{P} \boldsymbol{W}_{O_{(i,\cdot)}} \boldsymbol{W}_V(\boldsymbol{z}_{n(\cdot,s)} + \boldsymbol{\delta}_s^{(h)})$$
$$\cdot softmax((\boldsymbol{z}_{n(\cdot,s)} + \boldsymbol{\delta}_s^{(h)})^\top \boldsymbol{W}_K^\top \boldsymbol{W}_Q(\boldsymbol{z}_{n(\cdot,s)} + \boldsymbol{\delta}_l^{(h)}))), \tag{A29}$$

we have $\boldsymbol{W}_K = \beta_2 \cdot \boldsymbol{P}_x$, $\boldsymbol{W}_Q = \beta_2 \cdot \boldsymbol{I}$, and $\boldsymbol{W}_V = \boldsymbol{P}_x$ where $\beta_2 = \Theta(1)$. To avoid multiple superscripts, we use $\boldsymbol{\delta}$ to denote $\boldsymbol{\delta}^{(h)}$ since that $h$ is fixed in this proof. We use $\boldsymbol{\delta}^{(t)}$ to denote the update of $\boldsymbol{\delta}$ at $t$-th iteration. Then,

$$
\frac{\partial f_\theta(\boldsymbol{x}_n, \boldsymbol{\delta})}{\partial \boldsymbol{\delta}_j}
$$

$$
\begin{aligned}
= \sum_{l=1}^{P} \sum_{i=1}^{m} a_{l(i)} \mathbb{1}[\sum_{s=1}^{P} \boldsymbol{W}_{O_{(i,\cdot)}}(\boldsymbol{z}_{n(\cdot,P_{s,2})} + \boldsymbol{\delta}_{P_{s,2}}) \mathrm{softmax}((\boldsymbol{z}_{n(\cdot,P_{s,2})} + \boldsymbol{\delta}_{P_{s,2}})^\top (\boldsymbol{z}_{n(\cdot,s)} \\
+ \boldsymbol{\delta}_l)) \geq 0] \cdot \Big( \mathrm{softmax}((\boldsymbol{z}_{n(\cdot,P_{s,2})} + \boldsymbol{\delta}_{P_{s,2}})^\top (\boldsymbol{z}_{n(\cdot,s)} + \boldsymbol{\delta}_l)) \boldsymbol{W}_{O_{(i,\cdot)}} \\
+ \mathbb{1}[j \neq l] \boldsymbol{W}_{O_{(i,\cdot)}}(\boldsymbol{z}_{n(\cdot,j)} + \boldsymbol{\delta}_j) \cdot (\boldsymbol{z}_{n(\cdot,j)} + \boldsymbol{\delta}_l) \cdot (-\mathrm{softmax}(\beta_2^2(\boldsymbol{z}_{n(\cdot,j)} + \boldsymbol{\delta}_j)^\top \\
\cdot (\boldsymbol{z}_{n(\cdot,l)} + \boldsymbol{\delta}_l))) \mathrm{softmax}(\beta_2^2 (\boldsymbol{z}_{n(\cdot,l)} + \boldsymbol{\delta}_l)^\top (\boldsymbol{z}_{n(\cdot,l)} + \boldsymbol{\delta}_l)) \\
+ \mathbb{1}[j = l] \boldsymbol{W}_{O_{(i,\cdot)}}(\boldsymbol{z}_{n(\cdot,l)} + \boldsymbol{\delta}_l) \mathrm{softmax}(\beta_2^2 (\boldsymbol{z}_{n(\cdot,l)} + \boldsymbol{\delta}_l)^\top (\boldsymbol{z}_{n(\cdot,l)} + \boldsymbol{\delta}_l)) \\
\cdot (1 - \mathrm{softmax}(\beta_2^2 (\boldsymbol{z}_{n(\cdot,l)} + \boldsymbol{\delta}_l)^\top (\boldsymbol{z}_{n(\cdot,j)} + \boldsymbol{\delta}_l)))(\boldsymbol{z}_{n(\cdot,l)} + \boldsymbol{\delta}_l)
\end{aligned}
\tag{A30}
$$

Let $t = 0$. For $y^n = +1$, Note that if $\boldsymbol{z}_n = [\boldsymbol{e}_3, \boldsymbol{e}_3, \cdots, \boldsymbol{e}_3, \boldsymbol{e}_1, \boldsymbol{e}_3, \cdots, \boldsymbol{e}_3]$ without noise, the loss is 0. Hence, we compute the loss from $\boldsymbol{z}_n = [\boldsymbol{e}_4, \boldsymbol{e}_4, \cdots, \boldsymbol{e}_4, \boldsymbol{e}_1, \boldsymbol{e}_4, \cdots, \boldsymbol{e}_4]$.

$$
\mathbb{E}[\mathbb{1}[\sum_{s=1}^{P} \boldsymbol{W}_{O_{(i,\cdot)}}(\boldsymbol{x}_{n(\cdot,s)} + \boldsymbol{\delta}_s^{(t)}) \mathrm{softmax}(\beta_2^2(\boldsymbol{z}_{n(\cdot,P_{s,2})} + \boldsymbol{\delta}_{P_{s,2}}^{(t)})^\top (\boldsymbol{z}_{n(\cdot,l)} + \boldsymbol{\delta}_l^{(t)})) \geq 0]
$$

$$
= \mathrm{Pr}(\sum_{s=1}^{L} \boldsymbol{W}_{O_{(i,\cdot)}}(\boldsymbol{z}_{n(\cdot,P_{s,2})} + \boldsymbol{\delta}_{P_{s,2}}^{(t)}) \mathrm{softmax}(\beta_2^2(\boldsymbol{z}_{n(\cdot,P_{s,2})} + \boldsymbol{\delta}_{P_{s,2}}^{(t)})^\top (\boldsymbol{z}_{n(\cdot,l)} + \boldsymbol{\delta}_l^{(t)})) \geq 0)
\tag{A31}
$$

for $\boldsymbol{W}_{O_{(i,\cdot)}} = \boldsymbol{e}_1$ or $\boldsymbol{e}_4$. We can finally show that with a high probability, the above indicator is close to 1. Meanwhile, for $\boldsymbol{W}_{O_{(i,\cdot)}} = \boldsymbol{e}_2$ or $\boldsymbol{e}_3$, the indicator equals 0 or 1 with half probability when $t = 0$. Consider that $\boldsymbol{z}_{n(\cdot,j)}$ comes from $\boldsymbol{v}_4$, which means $\boldsymbol{z}_{n(\cdot,j)}$ is close to $\boldsymbol{v}_4$ by a noisy term. In this case, if $\boldsymbol{z}_{n(\cdot,l)}$ comes from $\boldsymbol{v}_1$,

$$
\mathrm{softmax}(\beta_2^2(\boldsymbol{z}_{n(\cdot,l)} + \boldsymbol{\delta}_l^{(t)})^\top (\boldsymbol{z}_{n(\cdot,l)} + \boldsymbol{\delta}_l^{(t)})) \geq \frac{1}{P}
\tag{A32}
$$

$$
\mathrm{softmax}(\beta_2^2(\boldsymbol{z}_{n(\cdot,j)} + \boldsymbol{\delta}_j)^\top (\boldsymbol{z}_{n(\cdot,l)} + \boldsymbol{\delta}_l^{(t)})) = \Theta(\frac{1}{P})
\tag{A33}
$$

If $\boldsymbol{z}_{n(\cdot,l)}$ comes from $\boldsymbol{v}_4$, then

$$
\mathrm{softmax}(\beta_2^2(\boldsymbol{z}_{n(\cdot,l)} + \boldsymbol{\delta}_l^{(t)})^\top (\boldsymbol{z}_{n(\cdot,l)} + \boldsymbol{\delta}_l^{(t)})) \geq \frac{1}{P}
\tag{A34}
$$

$$
\mathrm{softmax}(\beta_2^2(\boldsymbol{z}_{n(\cdot,j)} + \boldsymbol{\delta}_j^{(t)})^\top (\boldsymbol{z}_{n(\cdot,l)} + \boldsymbol{\delta}_l^{(t)})) = \Theta(\frac{1}{P})
\tag{A35}
$$

Then we consider that $\boldsymbol{z}_{n(\cdot,j)}$ comes from $\boldsymbol{e}_1$. In this case, if $\boldsymbol{z}_{n(\cdot,l)}$ comes from $\boldsymbol{v}_1$, then

$$
\mathrm{softmax}(\beta_2^2(\boldsymbol{z}_{n(\cdot,j)} + \boldsymbol{\delta}_j^{(t)})^\top (\boldsymbol{z}_{n(\cdot,l)} + \boldsymbol{\delta}_l^{(t)})) \geq \frac{1}{P}
\tag{A36}
$$

If $\boldsymbol{z}_{n(\cdot,l)}$ comes from $\boldsymbol{v}_4$,

$$
\mathrm{softmax}(\beta_2^2(\boldsymbol{z}_{n(\cdot,j)} + \boldsymbol{\delta}_j^{(t)})^\top (\boldsymbol{z}_{n(\cdot,l)} + \boldsymbol{\delta}_l^{(t)})) \leq \frac{1}{P}
\tag{A37}
$$

Therefore, if $\boldsymbol{z}_{n(\cdot,j)}$ comes from $\boldsymbol{v}_1$,

$$
\frac{\partial f_\theta(\boldsymbol{x}_n, \boldsymbol{\delta}^{(t)})}{\partial \boldsymbol{\delta}_j^{(t)}} = \frac{1}{4P} \lambda \boldsymbol{e}_1 + \Theta(\frac{1}{P})(-\boldsymbol{e}_2 + \boldsymbol{e}_3 - \boldsymbol{e}_4),
\tag{A38}
$$

and if $\boldsymbol{z}_{n(\cdot,j)}$ comes from $\boldsymbol{v}_4$,

$$
\frac{\partial f_\theta(\boldsymbol{x}_n, \boldsymbol{\delta}^{(t)})}{\partial \boldsymbol{\delta}_j^{(t)}} = -\frac{1}{4P} \lambda \boldsymbol{e}_4 + \Theta(\frac{1}{P})(-\boldsymbol{e}_2 + \boldsymbol{e}_3 + \boldsymbol{e}_1),
\tag{A39}
$$

where $\lambda = \mu = \Theta(1)$. The last terms in (A38) and (A39) come from the indicators from other $\boldsymbol{W}_O$ neurons, which may become 1 because of feature noises. Note that when $t \geq 2$, since the data which contains $\boldsymbol{e}_2$ and $\boldsymbol{e}_3$ would similarly contribute to the overall gradient, there will be a close amount of $\boldsymbol{e}_1$ and $\boldsymbol{e}_2$ in $\boldsymbol{\delta}_j^{(t)}$ and a close amount of $\boldsymbol{e}_3$ and $\boldsymbol{e}_4$ in $\boldsymbol{\delta}_j^{(t)}$. Hence, when $k\mu < \Theta(1)$,

$$
\mathbb{E}[\boldsymbol{\delta}_j^{(t)}] = \mathbb{E}[\boldsymbol{\delta}_j^{(0)}] - \mathbb{E}[\eta \sum_{b=1}^{t} \frac{1}{B} \sum_{n \in \mathcal{B}_b} \frac{\partial}{\partial \boldsymbol{\delta}_j} \ell(f_{\boldsymbol{\theta}}(\boldsymbol{x}_n, \boldsymbol{\delta}^{(b)}), y_n)]
$$

$$
= \eta t \frac{1}{4P} (\lambda \boldsymbol{e}_1 + \lambda \boldsymbol{e}_2 - \mu \boldsymbol{e}_3 - \mu \boldsymbol{e}_4)
$$

$$
= k(\lambda \boldsymbol{e}_1 + \lambda \boldsymbol{e}_2 - \mu \boldsymbol{e}_3 - \mu \boldsymbol{e}_4),
$$
(A40)

$$
\boldsymbol{\delta}_j^{(t)} = \mathbb{E}[\boldsymbol{\delta}_j^{(t)}] + \frac{\eta t}{L} \sqrt{\frac{\log Bt}{Bt}} (\pm \boldsymbol{e}_1 \pm \boldsymbol{e}_2 \pm \boldsymbol{e}_3 \pm \boldsymbol{e}_4)
$$
(A41)

where $\lambda \geq \Theta(1) \cdot (1 - \sigma P)$, $\mu \geq \Theta(1) \cdot (1 - \sigma P)$ for $t \geq 2$. The term $(1 - \sigma P)$ comes from that for $\boldsymbol{W}_{O_{(i,\cdot)}} = \boldsymbol{e}_1$ or $\boldsymbol{e}_4$,

$$
\mathbb{E}[\mathbb{1}[\sum_{s=1}^{P} \boldsymbol{W}_{O_{(i,\cdot)}}(\boldsymbol{z}_{n(\cdot, P_{s,2})} + \boldsymbol{\delta}_{P_{s,2}}^{(t)}) \text{softmax}(\beta_2^2 (\boldsymbol{z}_{n(\cdot, P_{s,2})} + \boldsymbol{\delta}_{P_{s,2}}^{(t)})^\top (\boldsymbol{z}_{n(\cdot, l)} + \boldsymbol{\delta}_l^{(t)})) \geq 0]
$$

$$
\geq 1 - e^{\frac{(Bt)^2}{\sigma^2 P^2}} \geq 1 - \sigma P
$$
(A42)

given $B \geq \Theta(1)$ by Hoeffding inequality. When $k\mu \geq \Theta(1)$, for $\boldsymbol{z}_n = [\boldsymbol{e}_4, \boldsymbol{e}_4, \cdots, \boldsymbol{e}_4, \boldsymbol{e}_1, \boldsymbol{e}_4, \cdots, \boldsymbol{e}_4]$,

$$
\boldsymbol{z}_{n(\cdot, j)} + \boldsymbol{\delta}_j^{(t)} = k\lambda(\boldsymbol{e}_1 + \boldsymbol{e}_2) - k\mu \boldsymbol{e}_3 + (1 - k\mu)\boldsymbol{e}_4
$$
(A43)

for $\boldsymbol{z}_{n(\cdot, j)}$ from $\boldsymbol{v}_4$. Then,

$$
\mathbb{E}[\mathbb{1}[\sum_{s=1}^{P} \boldsymbol{e}_1(\boldsymbol{z}_{n(\cdot, P_{s,2})} + \boldsymbol{\delta}_{P_{s,2}}^{(t)}) \text{softmax}(\beta_2^2 (\boldsymbol{z}_{n(\cdot, P_{s,2})} + \boldsymbol{\delta}_{P_{s,2}}^{(t)})^\top (\boldsymbol{z}_{n(\cdot, l)} + \boldsymbol{\delta}_l^{(t)}))]] \geq 1 - e^{\frac{(Bt)^2}{\sigma^2}} \geq 1 - \sigma
$$
(A44)

$$
\Pr(\sum_{s=1}^{P} \boldsymbol{e}_4(\boldsymbol{z}_{n(\cdot, P_{s,2})} + \boldsymbol{\delta}_{P_{s,2}}^{(t)}) \text{softmax}(\beta_2^2 (\boldsymbol{z}_{n(\cdot, P_{s,2})} + \boldsymbol{\delta}_{P_{s,2}}^{(t)})^\top (\boldsymbol{z}_{n(\cdot, l)} + \boldsymbol{\delta}_l^{(t)}))) \leq e^{-\frac{1}{\sigma^2}} \leq e^{-P^2}
$$
(A45)

Hence, with a probability at least $1 - e^{-P^2}$, no patches is activated by $\boldsymbol{e}_4$. For $\boldsymbol{z}_{n(\cdot, k)}$ from $\boldsymbol{v}_1$ and $\boldsymbol{z}_{n(\cdot, j)}$ from $\boldsymbol{v}_4$, we have

$$
\text{softmax}((\boldsymbol{z}_{n(\cdot, k)} + \boldsymbol{\delta}_k^{(t)})^\top (\boldsymbol{z}_{n(\cdot, k)} + \boldsymbol{\delta}_k^{(t)})) \geq \frac{1}{P}
$$
(A46)

$$
\text{softmax}((\boldsymbol{z}_{n(\cdot, j)} + \boldsymbol{\delta}_j^{(t)})^\top (\boldsymbol{z}_{n(\cdot, k)} + \boldsymbol{\delta}_k^{(t)})) = \Theta(\frac{1}{P})
$$
(A47)

$$
\text{softmax}((\boldsymbol{z}_{n(\cdot, j)} + \boldsymbol{\delta}_j^{(t)})^\top (\boldsymbol{z}_{n(\cdot, j)} + \boldsymbol{\delta}_j^{(t)})) \geq \frac{1}{P}
$$
(A48)

$$
\text{softmax}((\boldsymbol{z}_{n(\cdot, k)} + \boldsymbol{\delta}_k^{(t)})^\top (\boldsymbol{z}_{n(\cdot, j)} + \boldsymbol{\delta}_j^{(t)})) = \Theta(\frac{1}{P})
$$
(A49)

Therefore, when $k\mu > \Theta(1)$, i.e., $t \geq t_0 = 4P\eta^{-1}(1 - \sigma P)^{-1}$ we have

$$
\boldsymbol{\delta}_j^{(t)} = \mathbb{E}[\boldsymbol{\delta}_j^{(t)}] + \frac{\eta t}{P} \sqrt{\frac{\log B(t - t_0)}{B(t - t_0)}} (\pm(\boldsymbol{e}_1 + \boldsymbol{e}_2) \pm \frac{1}{P} e^{-P^4}(\boldsymbol{e}_3 + \boldsymbol{e}_4))
$$

$$
= \mathbb{E}[\boldsymbol{\delta}_j^{(t_0)}] - \mathbb{E}[\eta \sum_{b=t_0}^{t} \frac{1}{B} \sum_{n \in \mathcal{B}_b} \frac{\partial}{\partial \boldsymbol{\delta}_j} \ell(f_{\boldsymbol{\theta}}(\boldsymbol{x}_n, \boldsymbol{\delta}^{(b)}), y_n)] \pm \frac{\eta t}{P} \sqrt{\frac{\log B(t - t_0)}{B(t - t_0)}} (\boldsymbol{e}_1 + \boldsymbol{e}_2)
$$

$$
= \mathbb{E}[\boldsymbol{\delta}_j^{(t_0)}] + \frac{\eta(t - t_0)}{4P} (\lambda \boldsymbol{e}_1 + \lambda \boldsymbol{e}_2 + \mu \boldsymbol{e}_3 + \mu \boldsymbol{e}_4) \pm \frac{\eta t}{P} \sqrt{\frac{\log B(t - t_0)}{B(t - t_0)}} (\boldsymbol{e}_1 + \boldsymbol{e}_2),
$$
(A50)

where $\lambda \gtrsim (1-\sigma)^{-1}$. Then,

$$\left| \boldsymbol{e}_3^\top \mathbb{E}[\eta \sum_{b=t_0}^t \frac{1}{B} \sum_{n \in \mathcal{B}_b} \frac{\partial}{\partial \boldsymbol{\delta}} \ell(f_{\boldsymbol{\theta}}(\boldsymbol{x}_n, \boldsymbol{\delta}^{(b)}), y_n)] \right| \lesssim \eta(t-t_0)\frac{1}{P} \cdot \sqrt{\frac{\log B(t-t_0)}{B(t-t_0)}} \tag{A51}$$

$$\left| \boldsymbol{e}_4^\top \mathbb{E}[\eta \sum_{b=t_0}^t \frac{1}{B} \sum_{n \in \mathcal{B}_b} \frac{\partial}{\partial \boldsymbol{\delta}} \ell(f_{\boldsymbol{\theta}}(\boldsymbol{x}_n, \boldsymbol{\delta}^{(b)}), y_n)] \right| \lesssim \eta(t-t_0)\frac{1}{P} \cdot \sqrt{\frac{\log B(t-t_0)}{B(t-t_0)}} \tag{A52}$$

and thus $|\mu| \leq \Theta(1/\sqrt{B(t-t_0)})$. Hence, for $\boldsymbol{z}_{n(\cdot,k)}$ from $\boldsymbol{v}_1$ and $\boldsymbol{z}_{n(\cdot,j)}$ from $\boldsymbol{v}_4$,

$$(\boldsymbol{z}_{n(\cdot,k)} + \boldsymbol{\delta}_k^{(t)})^\top(\boldsymbol{z}_{n(\cdot,k)} + \boldsymbol{\delta}_k^{(t)}) - (\boldsymbol{z}_{n(\cdot,k)} + \boldsymbol{\delta}_k^{(t)})^\top(\boldsymbol{z}_{n(\cdot,j)} + \boldsymbol{\delta}_j^{(t)})$$
$$= \Theta(1) \cdot \frac{e^{\beta_2^2}}{e^{\beta_2^2} + P - 1}\left(\frac{e^{\beta_2^2}}{e^{\beta_2^2} + P - 1} + \boldsymbol{e}_1^\top \boldsymbol{\delta}^{(t)}\right) \tag{A53}$$

$$(\boldsymbol{z}_{n(\cdot,j)} + \boldsymbol{\delta}_j^{(t)})^\top(\boldsymbol{z}_{n(\cdot,k)} + \boldsymbol{\delta}_k^{(t)}) - (\boldsymbol{z}_{n(\cdot,j)} + \boldsymbol{\delta}_j^{(t)})^\top(\boldsymbol{z}_{n(\cdot,j)} + \boldsymbol{\delta}_j^{(t)})$$
$$= \Theta(1) \cdot \frac{e^{\beta_2^2}}{e^{\beta_2^2} + P - 1} \cdot \boldsymbol{e}_1^\top \boldsymbol{\delta}^{(t)} \tag{A54}$$

Since that $\beta_2 = \Theta(1)$, we have

$$\text{softmax}((\boldsymbol{z}_{n(\cdot,k)} + \boldsymbol{\delta}_k^{(t)})^\top(\boldsymbol{z}_{n(\cdot,k)} + \boldsymbol{\delta}_k^{(t)})) = \frac{e^{\Theta(1) \cdot \frac{\boldsymbol{e}_1^\top \boldsymbol{\delta}^{(t)}}{P}}}{P - 1 + e^{\Theta(1) \cdot \frac{\boldsymbol{e}_1^\top \boldsymbol{\delta}^{(t)}}{P}}} \tag{A55}$$

$$\text{softmax}((\boldsymbol{z}_{n(\cdot,k)} + \boldsymbol{\delta}_k^{(t)})^\top(\boldsymbol{z}_{n(\cdot,j)} + \boldsymbol{\delta}_j^{(t)})) = \frac{e^{\Theta(1) \cdot \frac{\boldsymbol{e}_1^\top \boldsymbol{\delta}^{(t)}}{P}}}{P - 1 + e^{\Theta(1) \cdot \frac{\boldsymbol{e}_1^\top \boldsymbol{\delta}^{(t)}}{P}}} \tag{A56}$$

To make

$$f_{\boldsymbol{\theta}}(\boldsymbol{x}_n, \boldsymbol{\delta}^{(t)}) \geq 1/P, \tag{A57}$$

we require that

$$\frac{e^{\Theta(1) \cdot \frac{\boldsymbol{e}_1^\top \boldsymbol{\delta}^{(t)}}{P}}}{P - 1 + e^{\Theta(1) \cdot \frac{\boldsymbol{e}_1^\top \boldsymbol{\delta}^{(t)}}{P}}} \cdot \frac{e^{\beta_2^2}}{e^{\beta_2^2} + P - 1} + \frac{P - 1}{P - 1 + e^{\Theta(1) \cdot \frac{\boldsymbol{e}_1^\top \boldsymbol{\delta}^{(t)}}{P}}} \cdot \frac{1}{e^{\beta_2^2}(P - 1) + 1} \geq \frac{1}{P} \tag{A58}$$

As a result, we finally need

$$e^{\Theta(1) \cdot \frac{\boldsymbol{e}_1^\top \boldsymbol{\delta}^{(t)}}{P}} \gtrsim P \tag{A59}$$

which holds as long as $t - t_0 \gtrsim P^2 \eta^{-1}(1-\sigma)^{-1} \log P$. Therefore, we have

$$f_{\boldsymbol{\theta}}(\boldsymbol{x}_n, \boldsymbol{\delta}) \geq 1/P \tag{A60}$$

for $\boldsymbol{x}_n$ that contains a patch from $\boldsymbol{v}_1$. We similarly have

$$f_{\boldsymbol{\theta}}(\boldsymbol{x}_n, \boldsymbol{\delta}) \leq -1/P \tag{A61}$$

for $\boldsymbol{x}_n$ that contains a patch from $\boldsymbol{v}_2$. To sum up, we need $t \geq \Theta(\eta^{-1}P^2(1-\sigma)^{-1} \log P)$ iterations.

### C.4.2 PROOF OF LEMMA 3

**Proof:**
To avoid multiple superscripts, we use $\boldsymbol{\delta}$ to denote $\boldsymbol{\delta}^{(h)}$ since that $h$ is fixed in this proof. We use $\boldsymbol{\delta}^{(t)}$ to denote the update of $\boldsymbol{\delta}$ at $t$-th iteration. For the network

$$f_{\boldsymbol{\theta}}(\boldsymbol{x}_n, \boldsymbol{\delta}) = \sum_{l=1}^P \sum_{i=1}^m a_{l(i)}^\top \text{Relu}(\sum_{j=1}^P \boldsymbol{W}_{O_{2(i,\cdot)}} \boldsymbol{W}_{V_2} \boldsymbol{z}_{n(\cdot,j)} \text{softmax}(\boldsymbol{z}_{n(\cdot,j)}^\top \boldsymbol{W}_{K_2}^\top \boldsymbol{W}_{Q_2} \boldsymbol{z}_{n(\cdot,l)})) \tag{A62}$$

where

$$\boldsymbol{z}_{n(\cdot,j)} = \text{Relu}(\sum_{s=1}^{P} \boldsymbol{W}_{O_1}\boldsymbol{W}_{V_1}(\boldsymbol{x}_{n(\cdot,P_{s,1})} + \boldsymbol{\delta}_s)\text{softmax}((\boldsymbol{x}_{n(\cdot,P_{s,1})} + \boldsymbol{\delta}_s)^{\top}\boldsymbol{W}_{K_1}^{\top}\boldsymbol{W}_{Q_1}(\boldsymbol{x}_j^n + \boldsymbol{\delta}_j))),$$

$$(A63)$$

we have

$$\frac{\partial f_{\boldsymbol{\theta}}(\boldsymbol{x}_n, \boldsymbol{\delta})}{\partial \boldsymbol{\delta}_s} = \sum_{j=1}^{P} \frac{\partial f_{\boldsymbol{\theta}}(\boldsymbol{x}_n, \boldsymbol{\delta})}{\partial \boldsymbol{z}_{n(\cdot,j)}} \frac{\partial \boldsymbol{z}_{n(\cdot,j)}}{\partial \boldsymbol{\delta}_s} \tag{A64}$$

Note that $\boldsymbol{W}_{Q_2} = \beta_2\boldsymbol{I}$, $\boldsymbol{W}_{Q_1} = \beta_1\boldsymbol{I}$, $\boldsymbol{W}_{K_2} = \beta_2\boldsymbol{P}_x$, $\boldsymbol{W}_{K_1} = \beta_1\boldsymbol{P}_x$,, $\boldsymbol{W}_{V_2} = \boldsymbol{P}_x$, $\boldsymbol{W}_{V_1} = \boldsymbol{P}_x$, where $\beta_1 = \Theta(1)$ and $\beta_2 = \Theta(1)$. Therefore,

$$\frac{\partial f_{\boldsymbol{\theta}}(\boldsymbol{x}_n, \boldsymbol{\delta})}{\partial \boldsymbol{z}_{n(\cdot,j)}}$$

$$= \sum_{l=1}^{P}\sum_{i=1}^{m} \boldsymbol{a}_{(l)_i}^{\top}\mathbb{1}[\sum_{s=1}^{P}\boldsymbol{W}_{O_{2(i,\cdot)}}\boldsymbol{z}_{n(\cdot,P_{s,2})}\text{softmax}(\beta_2^2\boldsymbol{z}_{n(\cdot,P_{s,2})}^{\top}\boldsymbol{z}_{n(\cdot,l)})]\Big(\text{softmax}(\beta_2^2\boldsymbol{z}_{n(\cdot,j)}^{\top}\boldsymbol{z}_{n(\cdot,l)})$$

$$\cdot \boldsymbol{W}_{O_{2(i,\cdot)}} + \mathbb{1}[j \neq l]\boldsymbol{W}_{O_{2(i,\cdot)}}\boldsymbol{z}_{n(\cdot,j)} \cdot \boldsymbol{z}_{n(\cdot,l)} \cdot (-\text{softmax}(\beta_2^2\boldsymbol{z}_{n(\cdot,j)}^{\top}\boldsymbol{z}_{n(\cdot,l)}))$$

$$\cdot \text{softmax}(\beta_2^2\boldsymbol{z}_{n(\cdot,l)}^{\top}\boldsymbol{z}_{n(\cdot,l)}) + \mathbb{1}[j = l]\boldsymbol{W}_{O_{2(i,\cdot)}}\boldsymbol{z}_{n(\cdot,l)}\text{softmax}(\beta_2^2\boldsymbol{z}_{n(\cdot,l)}^{\top}\boldsymbol{z}_{n(\cdot,l)})$$

$$\cdot (1 - \text{softmax}(\beta_2^2\boldsymbol{z}_{n(\cdot,l)}^{\top}\boldsymbol{z}_{n(\cdot,l)}))\boldsymbol{z}_{n(\cdot,l)}\Big)$$

$$(A65)$$

$$\frac{\partial \boldsymbol{z}_{n(\cdot,j)}}{\partial \boldsymbol{\delta}_k}$$

$$= \mathbb{1}[\sum_{s=1}^{P}\boldsymbol{W}_{O_1}(\boldsymbol{x}_{n(\cdot,P_{s,1})} + \boldsymbol{\delta}_s)\text{softmax}((\boldsymbol{x}_{n(\cdot,P_{s,1})} + \boldsymbol{\delta}_s)^{\top}(\boldsymbol{x}_j^n + \boldsymbol{\delta}_j))]\Big(\text{softmax}((\boldsymbol{x}_j^n + \boldsymbol{\delta}_j)^{\top}$$

$$\cdot (\boldsymbol{x}_{n(\cdot,l)} + \boldsymbol{\delta}_l))\boldsymbol{W}_{O_1} + \mathbb{1}[k \neq l]\boldsymbol{W}_{O_1}(\boldsymbol{x}_{n(\cdot,k)} + \boldsymbol{\delta}_k) \cdot (\boldsymbol{x}_{n(\cdot,l)} + \boldsymbol{\delta}_l)^{\top}$$

$$\cdot (-\text{softmax}(\beta_1^2(\boldsymbol{x}_j^n + \boldsymbol{\delta}_j)^{\top}(\boldsymbol{x}_{n(\cdot,l)} + \boldsymbol{\delta}_l)))\text{softmax}(\beta_1^2(\boldsymbol{x}_{n(\cdot,l)} + \boldsymbol{\delta}_l)^{\top}(\boldsymbol{x}_{n(\cdot,l)} + \boldsymbol{\delta}_l))$$

$$+ \mathbb{1}[k = l]\boldsymbol{W}_{O_1}(\boldsymbol{x}_{n(\cdot,l)} + \boldsymbol{\delta}_l)(\boldsymbol{x}_{n(\cdot,l)} + \boldsymbol{\delta}_l)^{\top}$$

$$\cdot \text{softmax}(\beta_1^2(\boldsymbol{x}_{n(\cdot,l)} + \boldsymbol{\delta}_l)^{\top}(\boldsymbol{x}_{n(\cdot,l)} + \boldsymbol{\delta}_l))$$

$$\cdot (1 - \text{softmax}(\beta_1^2(\boldsymbol{x}_{n(\cdot,l)} + \boldsymbol{\delta}_l)^{\top}(\boldsymbol{x}_{n(\cdot,l)} + \boldsymbol{\delta}_l)))\Big)$$

$$(A66)$$

Let $t = 0$. For $y^n = +1$, Note that if $\boldsymbol{x}_n = [\boldsymbol{e}_3, \boldsymbol{e}_3, \cdots, \boldsymbol{e}_3, \boldsymbol{e}_1, \boldsymbol{e}_3, \cdots, \boldsymbol{e}_3]$ without noise, the loss is 0. Hence, we compute the loss from $\boldsymbol{x}_n = [\boldsymbol{e}_4, \boldsymbol{e}_4, \cdots, \boldsymbol{e}_4, \boldsymbol{e}_1, \boldsymbol{e}_4, \cdots, \boldsymbol{e}_4]$.

$$\mathbb{E}[\mathbb{1}[\sum_{s=1}^{P}\boldsymbol{W}_{O_{(i,\cdot)}}(\boldsymbol{x}_{n(\cdot,P_{s,1})} + \boldsymbol{\delta}_{P_{s,1}}^{(t)})\text{softmax}(\beta_1^2(\boldsymbol{x}_{n(\cdot,P_{s,1})} + \boldsymbol{\delta}_{P_{s,1}}^{(t)})^{\top}(\boldsymbol{x}_{n(\cdot,l)} + \boldsymbol{\delta}_l)) \geq 0]$$

$$(A67)$$

$$= \Pr(\sum_{s=1}^{P}\boldsymbol{W}_{O_{(i,\cdot)}}(\boldsymbol{x}_{n(\cdot,P_{s,1})} + \boldsymbol{\delta}_{P_{s,1}}^{(t)})\text{softmax}(\beta_1^2(\boldsymbol{x}_{n(\cdot,P_{s,1})} + \boldsymbol{\delta}_{P_{s,1}}^{(t)})^{\top}(\boldsymbol{x}_{n(\cdot,l)} + \boldsymbol{\delta}_l)) \geq 0)$$

for $\boldsymbol{W}_{O_{(i,\cdot)}} = \boldsymbol{e}_1$ or $\boldsymbol{e}_4$. We can finally show that with a high probability, the above indicator is close to 1. Meanwhile, for $\boldsymbol{W}_{O_{(i,\cdot)}} = \boldsymbol{e}_2$ or $\boldsymbol{e}_3$, the indicator equals 0 or 1 with half probability when $t = 0$. Consider that $\boldsymbol{x}_{n(\cdot,j)}$ comes from $\boldsymbol{v}_4$. In this case, if $\boldsymbol{x}_{n(\cdot,l)}$ comes from $\boldsymbol{v}_1$,

$$\text{softmax}(\beta_1^2(\boldsymbol{x}_{n(\cdot,l)} + \boldsymbol{\delta}_l)^{\top}(\boldsymbol{x}_{n(\cdot,l)} + \boldsymbol{\delta}_l)) \geq \frac{1}{P} \tag{A68}$$

$$\text{softmax}(\beta_1^2(\boldsymbol{x}_j^n + \boldsymbol{\delta}_j^{(t)})^{\top}(\boldsymbol{x}_{n(\cdot,l)} + \boldsymbol{\delta}_l)) = \Theta(\frac{1}{P}) \tag{A69}$$

$$\text{softmax}(\beta_2^2\boldsymbol{z}_{n(\cdot,l)}^{\top}\boldsymbol{z}_{n(\cdot,l)}) \geq \frac{1}{P} \tag{A70}$$

$$\text{softmax}(\beta_2^2\boldsymbol{z}_{n(\cdot,j)}^{\top}\boldsymbol{z}_{n(\cdot,l)}) = \Theta(\frac{1}{P}) \tag{A71}$$

If $\boldsymbol{x}_{n(\cdot,l)}$ comes from $\boldsymbol{v}_4$, then

$$\text{softmax}(\beta_1^2(\boldsymbol{x}_{n(\cdot,l)} + \boldsymbol{\delta}_l^{(t)})^\top (\boldsymbol{x}_{n(\cdot,l)} + \boldsymbol{\delta}_l^{(t)})) \geq \frac{1}{P} \tag{A72}$$

$$\text{softmax}(\beta_1^2(\boldsymbol{x}_j^n + \boldsymbol{\delta}_j^{(t)})^\top (\boldsymbol{x}_{n(\cdot,l)} + \boldsymbol{\delta}_l^{(t)})) = \Theta(\frac{1}{P}) \tag{A73}$$

$$\text{softmax}(\beta_2^2 \boldsymbol{z}_{n(\cdot,l)}^\top \boldsymbol{z}_{n(\cdot,l)}) \geq \frac{1}{P} \tag{A74}$$

$$\text{softmax}(\beta_2^2 \boldsymbol{z}_{n(\cdot,j)}^\top \boldsymbol{z}_{n(\cdot,l)}) = \Theta(\frac{1}{P}) \tag{A75}$$

Then we consider that $\boldsymbol{x}_{n(\cdot,j)}$ comes from $\boldsymbol{v}_1$. In this case, if $\boldsymbol{z}_{n(\cdot,l)}$ comes from $\boldsymbol{v}_1$, then

$$\text{softmax}(\beta_1^2(\boldsymbol{x}_{n(\cdot,j)} + \boldsymbol{\delta}_j^{(t)})^\top (\boldsymbol{x}_{n(\cdot,l)} + \boldsymbol{\delta}_l^{(t)})) \geq \Theta(\frac{1}{P}) \tag{A76}$$

$$\text{softmax}(\beta_2^2 \boldsymbol{z}_{n(\cdot,j)}^\top \boldsymbol{z}_{n(\cdot,l)}) \geq \Theta(\frac{1}{P}) \tag{A77}$$

If $\boldsymbol{x}_{n(\cdot,l)}$ comes from $\boldsymbol{v}_4$,

$$\text{softmax}(\beta_1^2(\boldsymbol{x}_{n(\cdot,j)} + \boldsymbol{\delta}_j^{(t)})^\top (\boldsymbol{x}_{n(\cdot,l)} + \boldsymbol{\delta}_l^{(t)})) = \Theta(\frac{1}{P}) \tag{A78}$$

$$\text{softmax}(\beta_2^2 \boldsymbol{z}_{n(\cdot,j)}^\top \boldsymbol{z}_{n(\cdot,l)}) = \Theta(\frac{1}{P}) \tag{A79}$$

Therefore, if $\boldsymbol{x}_{n(\cdot,j)}$ comes from $\boldsymbol{v}_1$,

$$\frac{\partial f_{\boldsymbol{\theta}}(\boldsymbol{x}_n, \boldsymbol{\delta})}{\partial \boldsymbol{\delta}_j^{(t)}} = P \cdot \frac{1}{4P}\lambda(\boldsymbol{e}_1^\top \cdot \frac{1}{P}\boldsymbol{W}_{O_1})^\top = \frac{1}{4P}\boldsymbol{v}_1 + \Theta(\frac{1}{P})(-\boldsymbol{v}_2 + \boldsymbol{v}_3 - \boldsymbol{v}_4), \tag{A80}$$

and if $\boldsymbol{x}_{n(\cdot,j)}$ comes from $\boldsymbol{v}_4$,

$$\frac{\partial f_{\boldsymbol{\theta}}(\boldsymbol{x}_n, \boldsymbol{\delta})}{\partial \boldsymbol{\delta}_j^{(t)}} = -\frac{1}{4P}\mu\boldsymbol{v}_4 + \Theta(\frac{1}{P})(-\boldsymbol{v}_2 + \boldsymbol{v}_3 + \boldsymbol{v}_1), \tag{A81}$$

where $\lambda = \boldsymbol{\mu} = \Theta(1)$. Note that when $t \geq 2$, since the data which contains $\boldsymbol{v}_2$ and $\boldsymbol{v}_3$ would similarly contribute to the overall gradient, there will be a close amount of $\boldsymbol{v}_1$ and $\boldsymbol{v}_2$ in $\boldsymbol{\delta}_s^{(t)}$ and a close amount of $\boldsymbol{v}_3$ and $\boldsymbol{v}_4$ in $\boldsymbol{\delta}_s^{(t)}$. Hence, when $k\mu < \Theta(1)$,

$$\mathbb{E}[\boldsymbol{\delta}_s^{(t)}] = \mathbb{E}[\boldsymbol{\delta}_s^{(0)}] - \mathbb{E}[\eta \sum_{b=1}^t \frac{1}{B} \sum_{n \in \mathcal{B}_b} \frac{\partial}{\partial \boldsymbol{\delta}_s}\ell(f_{\boldsymbol{\theta}}(\boldsymbol{x}_n, \boldsymbol{\delta}_s^{(b)}), y_n)]$$
$$= \eta t \frac{1}{4P}(\lambda\boldsymbol{v}_1 + \lambda\boldsymbol{v}_2 - \mu\boldsymbol{v}_3 - \mu\boldsymbol{v}_4) \tag{A82}$$
$$= k(\lambda\boldsymbol{v}_1 + \lambda\boldsymbol{v}_2 - \mu\boldsymbol{v}_3 - \mu\boldsymbol{v}_4),$$

$$\boldsymbol{\delta}_s^{(t)} = \mathbb{E}[\boldsymbol{\delta}_s^{(t)}] + \frac{\eta t}{P}\sqrt{\frac{\log Bt}{Bt}}(\pm\boldsymbol{v}_1 \pm \boldsymbol{v}_2 \pm \boldsymbol{v}_3 \pm \boldsymbol{v}_4) \tag{A83}$$

where $\lambda \geq \Theta(1) \cdot (1 - \sigma P)$, $\mu \geq \Theta(1) \cdot (1 - \sigma P)$ for $t \geq 2$. The term $(1 - \sigma P)$ comes from that for $\boldsymbol{W}_{O_2(i,\cdot)} = \boldsymbol{v}_1$ or $\boldsymbol{v}_4$,

$$\mathbb{E}[\mathbb{1}[\sum_{s=1}^P \boldsymbol{W}_{O_1(i,\cdot)}(\boldsymbol{x}_{n(\cdot,P_s,1)} + \boldsymbol{\delta}_{P_s,1}^{(t)})\text{softmax}(\beta_1^2(\boldsymbol{x}_{n(\cdot,P_s,1)} + \boldsymbol{\delta}_{P_s,1}^{(t)})^\top (\boldsymbol{x}_{n(\cdot,l)} + \boldsymbol{\delta}_l^{(t)})) \geq 0]$$

$$\geq 1 - e^{\frac{(Bt)^2}{\sigma^2 P^2}} \geq 1 - \sigma P \tag{A84}$$

given $B \geq \Theta(1)$ by Hoeffding inequality. When $k\mu \geq \frac{\Theta(1)}{1+\gamma}$, we have that for $\boldsymbol{x}_{n(\cdot,j)}$ from $\boldsymbol{v}_4$,

$$
\mathbb{1}[\sum_{s=1}^{P} \boldsymbol{W}_{O_1}(\boldsymbol{x}_{n(\cdot,P_{s,1})} + \boldsymbol{\delta}_s)\text{softmax}(\beta_1^2(\boldsymbol{x}_{n(\cdot,P_{s,1})} + \boldsymbol{\delta}_s)^\top(\boldsymbol{x}_{n(\cdot,j)} + \boldsymbol{\delta}_j^{(t)})) \geq 0]
$$
$$
\geq [1, 1, -k\mu + (1 - k\mu)\gamma + \boldsymbol{v}_3^\top\boldsymbol{a}, -k\mu\gamma + 1 - k\mu + \boldsymbol{v}_4^\top\boldsymbol{a}]^\top
$$
$$
\geq [1, 1, 0, 0]^\top
$$

(A85)

where $\boldsymbol{a} \sim \mathcal{N}(0, \sigma^2\boldsymbol{I})$ in the first step, and the last step holds with probability at least

$$
\Pr(\boldsymbol{v}_4^\top\boldsymbol{a} - k\mu\gamma + 1 - k\mu \leq 0) \leq 1 - \Pr(\boldsymbol{v}_4^\top\boldsymbol{a} \geq \Theta(1)) \leq 1 - e^{\frac{1}{\sigma^2}} \leq 1 - e^{-P^2} \tag{A86}
$$

$$
\Pr(\boldsymbol{v}_3^\top\boldsymbol{a} - k\mu + (1 - k\mu)\gamma \leq 0) \leq 1 - \Pr(\boldsymbol{v}_3^\top\boldsymbol{a} \geq \Theta(1)) \leq 1 - e^{\frac{1}{\sigma^2}} \leq 1 - e^{-P^2} \tag{A87}
$$

Hence, for $\boldsymbol{x}_{n(\cdot,k)}$ from $\boldsymbol{v}_1$ and $\boldsymbol{x}_{n(\cdot,j)}$ from $\boldsymbol{v}_4$,

$$
(\boldsymbol{x}_{n(\cdot,k)} + \boldsymbol{\delta}_k^{(t)})^\top(\boldsymbol{x}_{n(\cdot,k)} + \boldsymbol{\delta}_k^{(t)}) - (\boldsymbol{x}_{n(\cdot,k)} + \boldsymbol{\delta}_k^{(t)})^\top(\boldsymbol{x}_{n(\cdot,j)} + \boldsymbol{\delta}_j^{(t)}) = \Theta(1) \cdot (1 + 2(k\mu)^2) \tag{A88}
$$

$$
(\boldsymbol{x}_{n(\cdot,j)} + \boldsymbol{\delta}_j^{(t)})^\top(\boldsymbol{x}_{n(\cdot,k)} + \boldsymbol{\delta}_k^{(t)}) - (\boldsymbol{x}_{n(\cdot,j)} + \boldsymbol{\delta}_j^{(t)})^\top(\boldsymbol{x}_{n(\cdot,j)} + \boldsymbol{\delta}_j^{(t)}) = \Theta(1) \cdot (2k\mu - 1) \tag{A89}
$$

Since that $\beta_1 = \Theta(1)$, we have

$$
\text{softmax}(\beta_1^2(\boldsymbol{x}_{n(\cdot,k)} + \boldsymbol{\delta}_k^{(t)})^\top(\boldsymbol{x}_{n(\cdot,k)} + \boldsymbol{\delta}_k^{(t)})) = \frac{e^{\Theta(1)\cdot(k\mu)^2}}{P - 1 + e^{\Theta(1)\cdot(k\mu)^2}} \tag{A90}
$$

$$
\text{softmax}(\beta_1^2(\boldsymbol{x}_{n(\cdot,k)} + \boldsymbol{\delta}_k^{(t)})^\top(\boldsymbol{x}_{n(\cdot,j)} + \boldsymbol{\delta}_j^{(t)})) = \frac{e^{\Theta(1)\cdot k\mu}}{P - 1 + e^{\Theta(1)\cdot k\mu}} \tag{A91}
$$

To make

$$
f_{\boldsymbol{\theta}}(\boldsymbol{x}_n, \boldsymbol{\delta}^{(t)}) \geq 1/P, \tag{A92}
$$

we require that

$$
\frac{e^{\Theta(1)\cdot(k\mu)^2}}{P - 1 + e^{\Theta(1)\cdot(k\mu)^2}} \cdot 1 \geq \frac{1}{P} \tag{A93}
$$

or

$$
\frac{e^{\Theta(1)\cdot k\mu}}{P - 1 + e^{\Theta(1)\cdot k\mu}} \cdot 1 \geq \frac{1}{P} \tag{A94}
$$

As a result, we finally need

$$
e^{\Theta(1)\cdot k\mu} \gtrsim 1 \tag{A95}
$$

which holds as long as $t \gtrsim P\eta^{-1}(1 - P\sigma)^{-1}(1 + \gamma)^{-1})$. With the same condition, we also have that for all $y^n = -1$,

$$
f_{\boldsymbol{\theta}}(\boldsymbol{x}_n, \boldsymbol{\delta}) \leq -1/P \tag{A96}
$$

To sum up, we need $t \geq \Theta(P\eta^{-1}(1 - P\sigma)^{-1}(1 + \gamma)^{-1}))$.

