# OpenReview forum: "Visual Prompting Reimagined: The Power of Activation Prompts"
_ICLR.cc/2025/Conference — ICLR 2025 Conference Withdrawn Submission_

### Official Review · Reviewer_V6EM · 2024-10-15

**Soundness:** 2
**Presentation:** 1
**Contribution:** 2
**Rating:** 3
**Confidence:** 4

**Summary:**

As visual prompting nowadays becomes a popular method to repurpose pretrained vision models for adaptation. The authors highlight a noticeable performance gap between VP and conventional fine-tuning methods. In this case, they introduce AP, extending the scope of (input-level) VP by enabling universal perturbations to be applied to activation maps with in the intermediate layers of the model.

**Strengths:**

The paper is easy to follow, the research topic is interesting, especially when considering the current storage in exploring prompt tuning.

The figures intuitively showcase the proposed method, and the findings are interesting.

**Weaknesses:**

1) The main experiments are conducted on ResNet-101 and ViT-Large/16, which are not the common practices, especially when considering AP is compared with VPT. In appendix, it is good to see that the authors report results on ViT-B/16 (Table A2). However, when looking into the performance itself, AP is not surprising to have a not satisfying results (i.e., with middle-level parameter usage and middle-level performance). (Also, here should be FGVC not FGCV). This further rise my questions on the contribution of AP (see 2).

2) The contribution/logic of this paper is poor, AP is more like a variant of VP in any sense. In Line 67-86, the authors discussed the difference between AP and VPT, as VPT applies prompt across multiple layers. In VPT paper page 19 sharing prompts, the authors clearly stated that they had initial exploration on weight sharing across layers. In this sense, AP is more like an observation-based variant of VPT. The discussion on VPT is still fundamental, however, further efficiency concerns might mislead the community (see 3).

3) In Line 245, the authors observed that ResNets and ViTs are exhibiting contrasting layer preferences for AP. During training, does that mean I should use grid search to go through all layers in order to find the best layer index? Figure 4 further proves my thoughts, as the layer index varies, the performance changes significantly, potentially leading to unstable training. The observations are interesting, however, the clear separation stated by the authors might mislead the prompt tuning research. In this sense, I do not think this paper is qualified for publication.

Suggestions: as the observations are interesting, the authors might think of completely reclaiming their current statement. Unfortunately, right now, I do not see fundamental changes/contributions at the structural level.

**Questions:**

Please see the Weaknesses.

---

### Official Review · Reviewer_V5tK · 2024-10-27

**Soundness:** 3
**Presentation:** 3
**Contribution:** 3
**Rating:** 6
**Confidence:** 4

**Summary:**

The paper, Visual Prompting Reimagined: The Power of Activation Prompts, proposes a novel approach called Activation Prompting (AP) to enhance Visual Prompting (VP) for adapting pretrained vision models to new tasks. Unlike traditional VP, which modifies the input data, AP applies perturbations to intermediate activation maps within the model, effectively broadening VP's scope. AP enables deeper customization by focusing prompts on specific model layers, allowing it to adapt based on model type and layer sensitivity.

**Strengths:**

AP introduces a approach to visual prompting by expanding from input-based modifications to activation-level prompts, allowing for targeted, layer-specific customization that enhances performance and efficiency. This technique appears to improve VP's effectiveness and better adapts it to different model architectures. Extensive evaluations across diverse datasets and models, including CNNs and ViTs, underscore AP’s adaptability and robustness, establishing its versatility for various vision tasks. Furthermore, the paper provides theoretical insights into layer-specific behavior, clarifying how AP preferences vary across model layers and types.

**Weaknesses:**

There are a few constrains for the work. 1. AP requires white-box access to the model's internal layers, limiting its applicability in scenarios where only black-box access is available, such as with proprietary models. 2. AP's effectiveness is not as pronounced in smaller models like ResNet-18 or ViT-Tiny, as noted in the paper. This limits its versatility, particularly for applications relying on compact or resource-constrained models.

**Questions:**

There are a few concerns regarding the submission: 1. The claim that PT is inferior to fine-tuning is not entirely accurate. This work "Facing the Elephant in the Room: Visual Prompt Tuning or Full Finetuning?" provides a systematic analysis of both techniques, and the choice between PT and FT generally depends on the specific task and model size; 2. More empirical analysis on computational latency is needed. Since AP requires a larger parameter budget than PT, what are the associated training costs?; 3. The paper shows promising few-shot learning performance, but the reliance on pretrained model size and specificity may pose overfitting risks, especially with limited data.

---

### Official Review · Reviewer_N9Ff · 2024-10-31

**Soundness:** 2
**Presentation:** 2
**Contribution:** 2
**Rating:** 3
**Confidence:** 5

**Summary:**

This paper introduced a generalized concept, called activation prompt (AP), which extends the scope of (input-level) VP by enabling universal perturbations to be applied to activation maps within the intermediate layers of the model.  The authors also showed that AP is closely related to normalized tuning in CNNs and ViTs. Experiments are conducted on 29 datasets to demonstrate the effectiveness of the proposed approach.

**Strengths:**

* This paper is easy to follow.

**Weaknesses:**

* The writing of this paper could be largely improved. The majority of the claims made by the authors are not supported. Statements from line 57 to line 65 are way too handwavy. These claims are not supported by any experiments or theories. From line 67 to line 86, the authors spent huge effort explaining the difference between VPT and the proposed AP which is very confusing. Overall, it is very confusing what the authors try to convey in the introduction section. It is usually expected to see certain background and motivation and the relation to the proposed approach.

* Limited novelty. Although the authors claimed that the proposed AP is very different from VPT, AP is essential identical to VPT, or in the authors words, a special case of VPT. AP is built on the claim that tradition VP only deals with input space, and AP deals also with intermediate features. Sadly, this claim is not true, since VPT-deep already studied adding prompts to intermediate features. That is to say, line 172-192 is already well studied by VPT.

* No performance gain compared to baselines. In table 4, the reported performance of the proposed AP outperforms none of the listed baselines. The comparison of efficiency is simply meaningless with degraded performance. To the extreme extent, updating nothing gives worst performance with best efficiency.

* Although the discussion of AP and normalization tuning shows something insights, this alone does not make much of a contribution.

**Questions:**

See weakness.

---

### Official Review · Reviewer_nQ5n · 2024-11-03

**Soundness:** 3
**Presentation:** 3
**Contribution:** 1
**Rating:** 3
**Confidence:** 5

**Summary:**

The authors propose a generalized approach to visual prompting (VP) by enabling learnable prompts to be added at deeper layers of the model. They also introduce a theoretical framework that explores the relationship between data sample complexity and the layer depth at which prompts are applied.

**Strengths:**

The paper is clearly written and easy to follow, with a well-structured presentation of the proposed ideas and theoretical analysis. The theory offers valuable insights into data complexity and the role of prompt application at different layers, which adds depth to the work.

**Weaknesses:**

- The primary concern is the validation of the core premise. The authors present AP (their approach) as a generalized or extended version of VP. However, AP and VP do not operate under the same assumptions: VP is typically applied in a black-box setting, while AP requires a white-box model, as noted in Line 65. **This distinction is critical, as it suggests that AP may be more aligned with Visual Prompt Tuning (VPT, Jia et al.), a classic white-box method.** In this respect, AP might actually be a specific case of VPT rather than a true generalization of VP. This discrepancy raises concerns about whether the connection between VP and AP has been overstated, potentially to differentiate it from existing VPT approaches. Consequently, the novelty of AP appears limited, as its distinctions from VPT are not substantial.

- The experimental validation of AP is also limited in comparison to VPT and related works, leaving questions about its empirical advantages.
- Additionally, while the theoretical contributions are interesting, the connection between the theory and the design of AP is not sufficiently strong.

**Questions:**

I believe this paper should be entirely rewritten and substantially revised.

---

### Note · Authors · 2024-11-16

I have read and agree with the venue's withdrawal policy on behalf of myself and my co-authors.